# Compactness and Consistency: A Conjoint Framework for Deep Graph Clustering

**Wei Ju**[1], **Siyu Yi**[*2], **Kangjie Zheng**[3], **Yifan Wang**[4], **Ziyue Qiao**[5], **Li Shen**[6], **Yongdao Zhou**[*7], **Xiaochun Cao**[6], **Jiancheng Lv**[1]

[1]College of Computer Science, Sichuan University
[2]College of Mathematics, Sichuan University
[3]Wellcome Sanger Institute [4]University of International Business and Economics
[5]School of Computing and Information Technology, Great Bay University
[6]School of Cyber Science and Technology, Shenzhen Campus of Sun Yat-sen University
[7]NITFID, School of Statistics and Data Science, Nankai University
`juwei@scu.edu.cn, siyuyi@scu.edu.cn, ydzhou@nankai.edu.cn`

## Abstract

Graph clustering is a fundamental task in data analysis, aiming at grouping nodes with similar characteristics in the graph into clusters. This problem has been widely explored using graph neural networks (GNNs) due to their ability to leverage node attributes and graph topology for effective cluster assignments. However, representations learned through GNNs typically struggle to capture global relationships between nodes via local message-passing mechanisms. Moreover, the redundancy and noise inherently present in graph data may easily result in node representations lacking compactness and robustness. To address these issues, we propose a conjoint framework CoCo, which captures compactness and consistency in the learned node representations for deep graph clustering. Technically, our CoCo leverages graph convolutional filters to learn robust node representations from both local and global views, and then encodes them into low-rank compact embeddings, thus effectively removing the redundancy and noise as well as uncovering the intrinsic underlying structure. To further enrich the node semantics, we develop a consistency learning strategy based on compact embeddings to facilitate knowledge transfer from the two perspectives. Our experimental results indicate that our CoCo outperforms state-of-the-art counterparts on various datasets. The code is available at https://github.com/juweipku/CoCo.

## 1 Introduction

Graph clustering, as a critical task in network analysis and machine learning, plays a significant role in organizing and understanding complex relational data (Liu et al., 2022b). It involves partitioning the nodes of a graph into clusters, aiming to group nodes that exhibit similar characteristics or share related patterns. Its application spans across various domains, including social network analysis, biological networks, and recommender systems, among others. By identifying cohesive groups of nodes, graph clustering provides valuable insights into the underlying structure and connections within the data, enabling more effective knowledge extraction and decision-making processes.

For the past decades, many efforts have been devoted to gaining a profound understanding of this fundamental problem, traditional methods typically rely on hand-crafted features (Yan et al., 2006) or graph partitioning algorithms (Ng et al., 2001; Vidal, 2011), which aim to project data samples into a low-dimensional space while incorporating constraints to ensure clear separation between the samples. However, the conventional training paradigm tends to yield unsatisfactory outcomes, as the limited model capacity restricts their potential, failing to fully exploit the abundant structural information contained in the graph. This has led us to study deep graph clustering, offering superior adaptability and expressive power by automatically extracting informative features from graphs (Bo et al., 2020; Ju et al., 2023; Liang et al., 2025b).

---

*Corresponding Authors.

Recently, graph neural networks (GNNs) have emerged as an effective approach, achieving remarkable success in capturing complex dependencies in graph-structured data and serving as a promising tool for deep graph clustering. Based on this strength, many researchers have explored the potential of GNNs for deep graph clustering (Liu et al., 2022a; Yang et al., 2023; 2024; Liu et al., 2024a). For example, SDCN (Bo et al., 2020) is the first to integrate structural information into deep clustering by bridging autoencoder representations with GCN layers through a delivery operator, while MAGI (Liu et al., 2024a) introduces a community-aware graph clustering framework that uses modularity maximization as a contrastive pretext task to uncover communities and mitigate semantic drift. In addition, DMGC-GTN (Wang et al., 2025a) explores a novel multi-modal graph clustering method that integrates structural and feature information via graph smoothing and transformer to exploit their complementarity. These GNN-based methods offer a data-driven way to learn node representations inherently capturing node attributes and the relational dependencies among neighboring nodes, potentially leading to more meaningful cluster assignments.

Despite the tremendous success of previous methods, there still exist some inherent limitations. ***First***, **existing graph clustering methods often struggle to effectively capture global relationships among nodes without any intervention** (Chen et al., 2020a). Since effective local message passing mechanisms (Gilmer et al., 2017) in GNNs typically propagate information for only a few layers, they often overlook long-range dependencies, limiting their ability to capture the underlying node distribution and resulting in suboptimal clustering. For instance, existing works (MAGI and DMGC-GTN) typically only perform local augmentation or random walks on the original graph, which fails to capture longer-range dependencies and consequently prevents clusters from accurately representing the underlying community structure. ***Second***, **the inherent redundancy and noise present in graph data poses challenges in learning compact and informative node representations** (Kang et al., 2019). Existing graph clustering methods often overlook the inherent redundancy and noise in data, which inevitably skews the training process and hinders the exploration of the underlying structure among data, thereby obscuring important relationships and patterns, and ultimately leading to less discriminative embeddings.

To address these challenges, we propose a conjoint framework capturing both **Co**mpactness and **Co**nsistency in the learned node representations, abbreviated as CoCo. Specifically, to explore both neighbor information and long-range relationships between nodes from local and global views respectively, our CoCo first leverages the graph convolutional filters to encode the node attributes and graph topology based on the original graph and graph diffusion matrix, thus effectively learning complementary node representations. Then, we encode node representations into low-rank compact embeddings, which learns the optimal low-dimensional subspace to characterize the intrinsic underlying structure, thereby fully eliminating redundancy and noise. To well facilitate the knowledge sharing of the compact embeddings from the two perspectives, we introduce a consistency learning strategy to encourage the model to produce consistent similarity distributions for each node, thus further enabling the learned node representations with richer semantics from both local and global information. Comprehensive experimental results across various graph datasets demonstrate the superior performance and effectiveness of our method over previous approaches.

## 2 METHODOLOGY

**Notations.** Let $\mathcal{G} = \{\mathcal{V}, \mathcal{E}, \mathbf{X}, \mathbf{A}\}$ denote an attribute graph of $N$ nodes, where $\mathcal{V} = \{v_1, \cdots, v_N\}$ represents the set of nodes and $\mathcal{E} \subseteq \mathcal{V} \times \mathcal{V}$ is the set of edges. Denote by $\mathbf{X} \in \mathbb{R}^{N \times F}$ the attribute matrix of all nodes, where $F$ is the dimension of attributes. $\mathbf{A} \in \{0, 1\}^{N \times N}$ is the adjacency matrix, where $A_{ij} = 1$ if $(v_i, v_j) \in \mathcal{E}$. The normalized adjacency matrix is denoted as $\tilde{\mathbf{A}} = \hat{\mathbf{D}}^{-1/2}\hat{\mathbf{A}}\hat{\mathbf{D}}^{-1/2}$, where $\hat{\mathbf{A}}$ equals to $\mathbf{A} + \mathbf{I}_N$ with added self-connections, and $\hat{\mathbf{D}}$ is the diagonal degree matrix with $\hat{D}_{ii} = \sum_{j=1}^{N} \hat{A}_{ij}$. Then the symmetric normalized graph Laplacian matrix is defined as $\tilde{\mathbf{L}} = \mathbf{I}_N - \tilde{\mathbf{A}}$.

**Deep Graph Clustering.** The task of deep graph clustering is to partition an unlabeled graph with $N$ nodes into $C$ disjoint clusters, denoted as $\{\mathcal{C}_1, \ldots, \mathcal{C}_C\}$ based on a well-trained node representation matrix $\mathbf{Z} \in \mathbb{R}^{N \times D}$. In general, a self-supervised loss is developed to guide the training process to learn informative node representations. Then, a clustering algorithm such as $K$-means, spectral clustering, or a neural-network clustering layer, is performed on the trained node representations to output the clustering results. In this section, we present a novel framework CoCo for deep graph clustering. The complete framework is depicted in Figure 1.

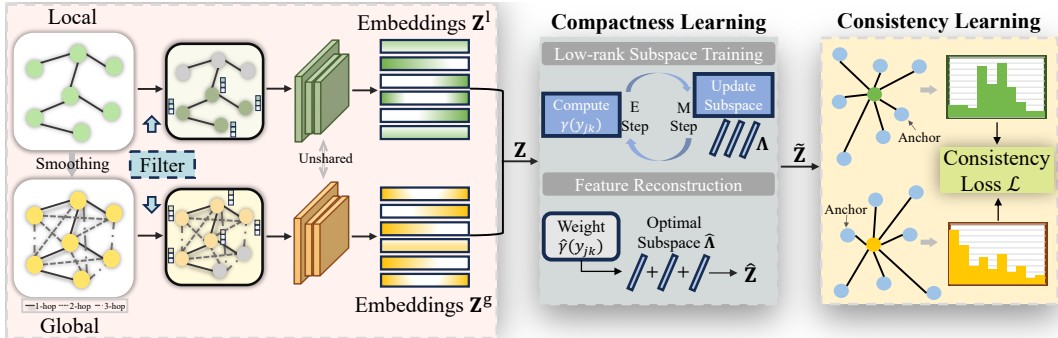

Figure 1: Illustration of the proposed framework CoCo.

## 2.1 Local- and Global-view Feature Extraction

To learn effective node representations, most existing methods employ graph convolution on the adjacency matrix, which propagates messages between one-hop neighbors. For capturing long-range neighbor information, the convolution layers are deepened, which inevitably leads to the over-smoothing issue, i.e., indistinguishable node representations of different clusters, due to the over-mixing of features and noises (Chen et al., 2020a). To alleviate this issue and fully explore graph topology, we leverage graph diffusion in this paper to smooth out the neighborhood over the graph. Formally, the graph diffusion matrix $\mathbf{S}$ is defined as: $\mathbf{S} = \alpha(\mathbf{I}_N - (1-\alpha)\tilde{\mathbf{A}})^{-1}$, which adopts the personalized PageRank (Page et al., 1999) with teleport probability $\alpha \in (0, 1)$. The elements in $\mathbf{S}$ measures the influence/correlation between all pairs of nodes. Compared with the adjacency matrix $\mathbf{A}$, $\mathbf{S}$ characterizes the soft relationships among the nodes, thus achieving the ability to globally exploit the long-range neighbor information. To reduce the computational complexity, there are fast approximations to achieves a linear runtime (Andersen et al., 2006; Wei et al., 2018) and we also sparsify (e.g., set values below a certain threshold to zero) $\mathbf{S}$ to obtain $\mathbf{S}'$, and modify it as $\hat{\mathbf{S}} \triangleq (\mathbf{S}' + \mathbf{S}'^{\top})/2$ to maintain symmetry. In the following, we treat the tuples $\{\mathbf{X}, \mathbf{A}\}$ and $\{\mathbf{X}, \hat{\mathbf{S}}\}$ as the *local* and *global* views to retrieve different knowledge over the given graph.

Inspired by Wu et al. (2019), the entanglement of graph convolutional filter and weight matrix in GNN will harm both the performance and robustness when learning node representations. Instead, we adopt the disentangled architecture to encode the attribute and structure information from local and global views. First, we utilize the generalized Laplacian smoothing filters to denoise the high-frequency components and integrate node attributes and structure information:

$$\tilde{\mathbf{X}}^l = (\mathbf{I}_N - \tilde{\mathbf{L}}^l/k^l)^t \mathbf{X}, \quad \tilde{\mathbf{X}}^g = (\mathbf{I}_N - \tilde{\mathbf{L}}^g/k^g)^t \mathbf{X}, \tag{1}$$

where $k^l, k^g (> 0)$ are real values and $t$ is the number of the filter layers. $\tilde{\mathbf{L}}^l (= \tilde{\mathbf{L}})$ and $\tilde{\mathbf{L}}^g$ are the local- and global-view normalized graph Laplacian matrix, in which $\tilde{\mathbf{L}}^g$ can be acquired like the expression of $\tilde{\mathbf{L}}$ except replacing $\mathbf{A}$ with $\mathbf{S}$. Theoretically, we can derive the following conclusion and the proof is shown in Appendix B.1.

> **Theorem 1.** $k^l = \tilde{\lambda}^l_{max}$ and $k^g = \tilde{\lambda}^g_{max}$ are the optimal choice to pursue low-pass filters, where $\tilde{\lambda}^g_{max}$ and $\tilde{\lambda}^l_{max}$ are the maximal eigenvalues of $\tilde{\mathbf{L}}^l$ and $\tilde{\mathbf{L}}^g$, respectively.

Then, to make node representations trainable, we learn weight parameters by feeding the filtered features into two unshared multi-layer perceptrons (MLP$_1$ and MLP$_2$) for two views as:

$$\mathbf{Z}^l = \text{MLP}_1(\tilde{\mathbf{X}}^l), \quad \mathbf{Z}^g = \text{MLP}_2(\tilde{\mathbf{X}}^g). \tag{2}$$

## 2.2 Compactness Learning for Redundancy Elimination

Low-rank representations can effectively exploit the inherent underlying correlation structure among data and suppress the impact of noise under the assumption that high-dimensional data points often intrinsically lie on a low-dimensional subspace (Ren et al., 2019; Tang et al., 2019; Jin et al., 2022a). For self-supervised learning, to achieve better fusion of the semantics from the local and global

views, we expect to utilize the same set of low-dimensional subspace to abstract and reconstruct the low-rank node representations of the two perspectives, which closes the gap between their semantic spaces and eliminates the redundancies in features, thereby uncovering the underlying data structure and yielding compact node representations from the two views.

**Low-rank Subspace Training.** Technically, we leverage the Gaussian mixture model (GMM, in Appendix A) (Richardson & Green, 1997) to learn the optimal low-dimensional subspace that can represent the local- and global-view node embeddings in Equation 2. Let $\bar{N} = 2N$, denote the embedding space as $\mathbf{Z} = (\mathbf{Z}^{\text{l}\top}, \mathbf{Z}^{\text{g}\top})^{\top} \in \mathbb{R}^{\bar{N} \times D}$, the learnable subspace as $\boldsymbol{\Lambda} \in \mathbb{R}^{\bar{N} \times K}$ ($K \ll D$), the $j$-th column of $\mathbf{Z}$ as $\mathbf{z}_{\cdot,j} = (z_{1j}, \ldots, z_{\bar{N}j})^{\top}$, and the $k$-th column of $\boldsymbol{\Lambda}$ as $\boldsymbol{\lambda}_{\cdot,k} = (\lambda_{1k}, \ldots, \lambda_{\bar{N}k})^{\top}$. We introduce the latent variable matrix $\mathbf{Y} \in \mathbb{R}^{D \times K}$, where the element $y_{jk}$ in $\mathbf{Y}$ indicates whether $\mathbf{z}_{\cdot,j}$ is related to $\boldsymbol{\lambda}_{\cdot,k}$. Under GMM, the goal is to maximize $\log p(\mathbf{Z}|\boldsymbol{\lambda})$ and the likelihood functions for the observed data $\mathbf{Z}$ and the complete data $\{\mathbf{Z}, \mathbf{Y}\}$ are proportionally formulated as:

$$p(\mathbf{Z}|\boldsymbol{\lambda}) \propto \prod_{j=1}^{D} \left( \sum_{k=1}^{K} \mathcal{N}(\mathbf{z}_{\cdot,j}|\boldsymbol{\lambda}_{\cdot,k}, \sigma\mathbf{I}_{\bar{N}}) \right), \ p(\mathbf{Z}, \mathbf{Y}|\boldsymbol{\lambda}) \propto \prod_{j=1}^{D} \prod_{k=1}^{K} \mathcal{N}(\mathbf{z}_{\cdot,j}|\boldsymbol{\lambda}_{\cdot,k}, \sigma\mathbf{I}_{\bar{N}})^{y_{jk}},$$

where $\sigma$ is a hyper-parameter to adjust the normal distribution. By implementing the expectation-maximization (EM) algorithm (Dempster et al., 1977), introduced in Appendix A , we can obtain in the E step that the posterior probability of the latent variable ($p(\mathbf{Y}|\mathbf{Z}, \boldsymbol{\lambda})$) is calculated by:

$$\gamma(y_{jk}) = p(y_{jk} = 1|\mathbf{z}_{\cdot,j}, \boldsymbol{\lambda}_{\cdot,k}^{\text{old}}) = \frac{\mathcal{N}(\mathbf{z}_{\cdot,j}|\boldsymbol{\lambda}_{\cdot,k}^{\text{old}}, \sigma\mathbf{I}_{\bar{N}})}{\sum_{k'=1}^{K} \mathcal{N}(\mathbf{z}_{\cdot,j}|\boldsymbol{\lambda}_{\cdot,k'}^{\text{old}}, \sigma\mathbf{I}_{\bar{N}})};$$

and the posterior expectation $Q(\boldsymbol{\lambda}, \boldsymbol{\lambda}^{\text{old}})$ is: $Q(\boldsymbol{\lambda}, \boldsymbol{\lambda}^{\text{old}}) = E_{\mathbf{Y}|\mathbf{Z}, \boldsymbol{\lambda}^{\text{old}}}(\log p(\mathbf{Z}, \mathbf{Y}|\boldsymbol{\lambda}))$. In the M step, we maximize $Q(\boldsymbol{\lambda}, \boldsymbol{\lambda}^{\text{old}})$ and the subspace is updated by:

$$\lambda_{ik}^{\text{new}} = \frac{1}{\sum_{j=1}^{D} \gamma(y_{jk})} \sum_{j=1}^{D} \gamma(y_{jk}) z_{ij}. \tag{3}$$

Here, the qualities of the Gaussian means $\boldsymbol{\lambda}_{\cdot,k}$ and the posteriors $\gamma(y_{jk})$ are critical. To simplify the model, we fix the mixture weights (priors) to be equal and the covariance matrices to be isotropic. It does not compromise the model's generality, since the equal prior does not affect the posterior trends, which are primarily data-driven. It also has advantages: (1) equal weights mitigate cluster collapse and promotes uniform coverage of the embedding space; and (2) fixing the covariance focuses the model's fitting capacity on the "mean-defined subspace", since based on the negative ELBO bound, maximizing the log-likelihood in the M-step is equivalent to minimizing the weighted squared-distance objective $\sum_{j,k} \gamma(y_{jk})\|\mathbf{z}_{\cdot,j} - \boldsymbol{\lambda}_{\cdot,k}\|^{2}$. Theoretically, the iterative algorithm is guaranteed to converge (Dempster et al., 1977), as stated in Remark 1 and proven in Appendix B.2. In the experiment, we verify that the subspace searching algorithm can achieve good performance within 10 iterations across different datasets, incurring negligible additional computational cost.

**Remark 1.** *In each iteration, we have* $\log p(\mathbf{Z}|\boldsymbol{\lambda}^{\text{new}}) \geq \log p(\mathbf{Z}|\boldsymbol{\lambda}^{\text{old}})$.

By repeatedly iterating the E step and M step until converging, the algorithm enforces the trained subspace $\boldsymbol{\Lambda}$ to effectively represent the core characteristics of the embedded representation $\mathbf{Z}$, as only the principal and cluster-level directions captured by the GMM means are retained while correlated or weakly informative variations are removed. As such, the intrinsic data relationship is preserved in $\boldsymbol{\Lambda}$ while removing the redundancy. Discussion on the comparison with other rank reduction ways can be found in Appendix C.

**Feature Reconstruction.** Further, to hold the main energy of the "clean" data (Ren et al., 2019), we perform data reconstruction to produce low-rank and compact representations stripped of redundancy. Concretely, we use the well-trained posterior probability of the latent variable $\hat{\gamma}(y_{jk})$ and the optimal subspace $\hat{\boldsymbol{\Lambda}} = (\hat{\lambda}_{ik})$ to linearly reconstruct the original features, i.e., each entry in the reconstructed embedding $\hat{\mathbf{Z}} = (\hat{z}_{ij}) \in \mathbb{R}^{\bar{N} \times D}$ is formulated as:

$$\hat{z}_{ij} = \sum_{k=1}^{K} \hat{\lambda}_{ik} \hat{\gamma}(y_{jk}). \tag{4}$$

Since the noise or unstable fluctuations in the original embeddings cannot be expressed within the constrained subspace and thus vanish when reconstructing onto the original dimensions (mathematically, we have $\text{rank}(\hat{\mathbf{Z}}) = \text{rank}(\hat{\boldsymbol{\Lambda}}\hat{\boldsymbol{\Gamma}}^{\top}) \leq \min\{\text{rank}(\hat{\boldsymbol{\Lambda}}), \text{rank}(\hat{\boldsymbol{\Gamma}})\} = K$, which indicates that $\hat{\mathbf{Z}}$

maintains the low-rank property). Further, we argue in Theorem 2 that the proposed approach reconstructs embedding $\hat{\mathbf{Z}}$ in a way that optimally preserves individual information and total variation of the original embedding $\mathbf{Z}$. The proof is shown in Appendix B.3.

---

**Theorem 2.** *Under the low-rank feature reconstruction defined in Equation 4, the following two conservation properties hold:*

*(1) **Individual Mass Conservation**: the aggregated information for each individual is preserved:*

$$\sum_{j=1}^{D} z_{ij} = \sum_{j=1}^{D} \hat{z}_{ij} \quad \text{for all } i \in \{1, \ldots, \bar{N}\}.$$

*(2) **Maximal Variation Preservation**: the reconstruction $\hat{\mathbf{Z}}$ maximally preserves the total variation of the original embedding $\mathbf{Z}$ among all low-rank factorizations. Specifically, it is the solution to the optimization problem:*

$$\hat{\mathbf{Z}} = \arg\min_{\mathbf{Z}' \in \mathbb{R}^{\bar{N} \times D}} \sum_{i=1}^{\bar{N}} \sum_{j=1}^{D} z_{ij}(z_{ij} - z'_{ij}) \quad \text{subject to } \mathbf{Z}' = \mathbf{\Lambda}\mathbf{\Gamma}^{\top},$$

*where $\mathbf{\Lambda} \in \mathbb{R}^{\bar{N} \times K}$ and $\mathbf{\Gamma} \in \mathbb{R}^{D \times K}$ with $K \ll D$.*

---

Theorem 2 (1) guarantees the invariance of each individual's total signal mass. This is crucial for fairness and interpretability, as it prevents the model from systematically biasing the reconstructed profiles of any individual; while Theorem 2 (2) ensures that our reconstruction prioritizes the retention of the significant variations (with large $z_{ij}$) due to the cross-term $\sum_{i,j} z_{ij} z'_{ij}$. This enables our low-rank reconstruction to align strongly with these dominant patterns. By leveraging the optimal low-dimensional subspace to reconstruct both the local- and global-view node embeddings, the semantic gap between them is also well alleviated. The obtained compact representation from both views can express the underlying data structure better, which is promising and beneficial to enhance the graph clustering. However, the above implementation is performed outside the gradient flow, we hence inject the tuned representations back into the gradient path using a residual connection:

$$\tilde{\mathbf{Z}} = (\tilde{\mathbf{Z}}^{\mathrm{l}\top}, \tilde{\mathbf{Z}}^{\mathrm{g}\top})^{\top} = \hat{\mathbf{Z}} + \mathbf{Z}. \tag{5}$$

On the one hand, it ensures the model remains trainable by allowing gradients to flow through the residual path; on the other hand, it combines the global trends captured by the low-rank component with the local details preserved in the original $\mathbf{Z}$, mitigating the over-smoothing that may result from relying solely on a low-rank constraint and preventing model collapse (He et al., 2016).

## 2.3 CONSISTENCY LEARNING FOR SEMANTIC ENHANCING

On the basis of compact node representations $\tilde{\mathbf{Z}}^{\mathrm{l}} = (\tilde{\mathbf{z}}_1^{\mathrm{l}}, \ldots, \tilde{\mathbf{z}}_N^{\mathrm{l}})^{\top}$ and $\tilde{\mathbf{Z}}^{\mathrm{g}} = (\tilde{\mathbf{z}}_1^{\mathrm{g}}, \ldots, \tilde{\mathbf{z}}_N^{\mathrm{g}})^{\top}$ in Equation 5, we develop the consistency learning to facilitate the exchange of knowledge between the two complementary perspectives. Meanwhile, we expect that the final representations can finely reflect the inherent relationships among nodes, thereby enhancing the label-free graph clustering. Toward this end, we share the compact semantics by comparing the similarities of each node to other samples in the embedding spaces of the two views.

We first randomly select a set of nodes over the given graph with indices $\{a_1, \ldots, a_M\}$ as the anchor samples. Then, we calculate the cosine similarities between each node and these anchor samples and formulate the similarity distribution by the softmax operation. Mathematically, for the $i$-th node representations from the local and global views, the similarity scores of the $m$-th anchor are:

$$p_m^i = \frac{\exp(\cos(\tilde{\mathbf{z}}_i^{\mathrm{l}}, \tilde{\mathbf{z}}_{a_m}^{\mathrm{l}})/\tau)}{\sum_{m'=1}^{M} \exp(\cos(\tilde{\mathbf{z}}_i^{\mathrm{l}}, \tilde{\mathbf{z}}_{a_m'}^{\mathrm{l}})/\tau)}, \quad q_m^i = \frac{\exp(\cos(\tilde{\mathbf{z}}_i^{\mathrm{g}}, \tilde{\mathbf{z}}_{a_m}^{\mathrm{g}})/\tau)}{\sum_{m'=1}^{M} \exp(\cos(\tilde{\mathbf{z}}_i^{\mathrm{g}}, \tilde{\mathbf{z}}_{a_m'}^{\mathrm{g}})/\tau)},$$

where $\cos(\mathbf{a}, \mathbf{b}) = \mathbf{a}^{\top}\mathbf{b}/(\|\mathbf{a}\| \cdot \|\mathbf{b}\|)$ is the cosine similarity, $\tau$ denotes the temperature parameter. For a comprehensive similarity measure, we need a large number of anchor samples so that they have large variations to cover the neighborhood of any node. However, it requires high computational costs to process too many samples in a single iteration. To address this problem, we maintain a memory bank with size $M$ as a queue defined on the fly by random nodes and calculate the similarity scores for each node and the samples in the queue. By dynamically updating the queue, we improve the diversity of the anchors with low complexity.

Table 1: Clustering performance on five benchmark datasets (mean ± standard deviation). The top two results for each method are marked in **bold** and underline, respectively.

| Dataset | Metric | SDCN | DFCN | AutoSSL | AFGRL | GDCL | ProGCL | CCGC | GraphLearner | MAGI | CoCo (Ours) |
|---|---|---|---|---|---|---|---|---|---|---|---|
| Cora | ACC | 35.60±2.83 | 36.33±0.49 | 63.81±0.57 | 26.25±1.24 | 70.83±0.47 | 57.13±1.23 | 73.88±1.20 | 74.91±1.78 | 76.21±0.50 | **79.36+0.69** |
| | NMI | 14.28±1.91 | 19.36±0.87 | 47.62±0.45 | 12.36±1.54 | 56.30±0.36 | 41.02±1.34 | 56.45±1.04 | 58.16±0.83 | 59.84±0.43 | **60.71±0.59** |
| | ARI | 07.78±3.24 | 04.67±2.10 | 38.92±0.77 | 14.32±1.87 | 48.05±0.72 | 30.71±2.70 | 52.51±1.89 | 53.82±2.25 | 57.63±0.81 | **58.76±1.47** |
| | F1 | 24.37±1.04 | 26.16±0.50 | 56.42±0.21 | 30.20±1.15 | 52.88±0.97 | 45.68±1.29 | 70.98±2.79 | 73.33±1.86 | 74.07±0.45 | **77.95±0.72** |
| AMAP | ACC | 53.44±0.81 | 76.82±0.23 | 54.55±0.97 | 75.51±0.77 | 43.75±0.78 | 51.53±0.38 | 77.25±0.41 | 77.24±0.87 | 75.42±3.22 | **79.27±0.70** |
| | NMI | 44.85±0.83 | 66.23±1.21 | 48.56±0.71 | 64.05±0.15 | 37.32±0.28 | 39.56±0.39 | 67.44±0.48 | 67.12±0.92 | 64.98±1.92 | **68.85±1.55** |
| | ARI | 31.21±1.23 | 58.28±0.74 | 26.87±0.34 | 54.45±0.48 | 21.57±0.51 | 34.18±0.89 | 57.99±0.66 | 58.14±0.82 | 55.68±2.88 | **60.94±1.51** |
| | F1 | 50.66±1.49 | 71.25±0.31 | 54.47±0.83 | 69.99±0.34 | 38.37±0.29 | 31.97±0.44 | 72.18±0.57 | 73.02±2.34 | **73.03±3.30** | 72.36±1.15 |
| BAT | ACC | 53.05±4.63 | 55.73±0.06 | 42.43±0.47 | 50.92±0.44 | 45.42±0.54 | 55.73±0.79 | 75.04±1.78 | 75.50±0.87 | 59.54±3.90 | **78.85±0.91** |
| | NMI | 25.74±5.71 | 48.77±0.51 | 17.84±0.98 | 27.55±0.62 | 31.70±0.42 | 28.69±0.92 | 50.23±2.43 | 50.58±0.90 | 29.83±5.13 | **55.00±0.87** |
| | ARI | 21.04±4.97 | 37.76±0.23 | 13.11±0.81 | 21.89±0.74 | 19.33±0.57 | 21.84±1.34 | 46.95±3.09 | 47.45±1.53 | 23.91±3.76 | **53.52±1.15** |
| | F1 | 46.45±5.90 | 50.90±0.12 | 34.84±0.15 | 46.53±0.57 | 39.94±0.57 | 56.08±0.89 | 74.90±1.80 | 75.40±0.88 | 59.12±6.11 | **78.56±1.01** |
| EAT | ACC | 39.07±1.51 | 49.37±0.19 | 31.33±0.52 | 37.42±1.24 | 33.46±0.18 | 43.36±0.87 | 57.19±0.66 | 57.22±0.73 | 49.10±1.50 | **58.87±0.49** |
| | NMI | 08.83±2.54 | 32.90±0.41 | 07.63±0.85 | 11.44±1.41 | 13.22±0.33 | 23.93±0.45 | 33.85±0.87 | 33.47±0.34 | 27.00±2.65 | **34.10±1.26** |
| | ARI | 06.31±1.95 | 23.25±0.18 | 02.13±0.67 | 06.57±1.73 | 04.31±0.29 | 15.03±0.98 | 27.71±0.41 | 26.21±0.81 | 21.52±1.01 | **27.91±1.52** |
| | F1 | 33.42±3.10 | 42.95±0.04 | 21.82±0.98 | 30.53±1.47 | 25.02±0.21 | 42.54±0.45 | 57.09±0.94 | 57.53±0.67 | 44.38±2.20 | **58.06±2.64** |
| UAT | ACC | 52.25±1.91 | 33.61±0.09 | 42.52±0.64 | 41.50±0..25 | 48.70±0.06 | 45.38±0.58 | 56.34±1.11 | 55.31±2.42 | 50.35±0.16 | **59.68±0.36** |
| | NMI | 21.61±1.26 | 26.49±0.41 | 17.86±0.22 | 17.33±0.54 | 25.10±0.01 | 22.04±2.23 | 28.15±1.92 | 24.40±1.69 | 21.45±0.28 | **30.12±0.51** |
| | ARI | 21.63±1.49 | 11.87±0.23 | 13.13±0.71 | 13.62±0.57 | 21.76±0.01 | 14.74±1.99 | 25.52±2.09 | 22.14±1.67 | 17.79±0.23 | **29.46±0.47** |
| | F1 | 45.59±3.54 | 25.79±0.29 | 34.94±0.87 | 36.52±0.89 | 45.69±0.08 | 39.30±1.82 | 55.24±1.69 | 52.77±2.61 | 47.52±0.13 | **58.03±0.34** |

With the local- and global-view similarity distributions $\mathbf{p}^i = (p_1^i, \ldots, p_M^i)$ and $\mathbf{q}^i = (q_1^i, \ldots, q_M^i)$, we encourage the consistency of them to facilitate the knowledge transfer and mutually enhance the representation semantics. Formally, we define the consistency learning loss as:

$$\mathcal{L} = \frac{1}{2N} \sum_{i=1}^{N} \left( \mathrm{KL}(\mathbf{p}^i \| \mathbf{q}^i) + \mathrm{KL}(\mathbf{q}^i \| \mathbf{p}^i) \right), \tag{6}$$

where $\mathrm{KL}(\cdot \| \cdot)$ is the Kullback-Leibler (KL) divergence. In the training, we minimize $\mathcal{L}$ to optimize our proposed CoCo and enhance self-supervised learning. After converging, we fuse the local- and global-view representations by:

$$\mathbf{Z}^{\mathrm{F}} = (\tilde{\mathbf{Z}}^{\mathrm{l}} + \tilde{\mathbf{Z}}^{\mathrm{g}})/2. \tag{7}$$

Then, we perform $K$-means on the fused node representation $\mathbf{Z}^{\mathrm{F}}$ to obtain the clustering results. An outline of the training procedure is provided in Appendix D. A detailed analysis of time and space complexities can be found in Section 3.8 and Appendix E.

## 3 EXPERIMENT

### 3.1 EXPERIMENTAL SETUP

We evaluate our CoCo with five widely used benchmark datasets for deep graph clustering, i.e., Cora (Sen et al., 2008), AMAP (Shchur et al., 2018), BAT (Liu et al., 2023c), EAT (Liu et al., 2023c), and UAT (Liu et al., 2023c). To comprehensively assess the effectiveness of our CoCo, we benchmark it against leading state-of-the-art methods, including *antoencoder-based methods*, i.e., DEC (Xie et al., 2016), IDEC (Guo et al., 2017), DAEGC (Wang et al., 2019), ARGA (Pan et al., 2019), SDCN (Bo et al., 2020), DFCN (Tu et al., 2021), and *contrastive learning-based methods*, i.e., AGE (Cui et al., 2020), MVGRL (Hassani & Khasahmadi, 2020), GDCL (Zhao et al., 2021), AutoSSL (Jin et al., 2022b), AGC-DRR (Gong et al., 2022), AFGRL (Lee et al., 2022), GDCL (Zhao et al., 2021), ProGCL (Xia et al., 2022), RGC (Liu et al., 2023a), Dink-Net (Liu et al., 2023b), CCGC (Yang et al., 2023), GraphLearner (Yang et al., 2024), and MAGI (Liu et al., 2024a). Details on evaluation metrics and implementation are presented in Appendix F.

### 3.2 EXPERIMENTAL RESULTS

In Table 1 and Appendix G, we present the quantitative results of our proposed CoCo, compared with various competitive deep graph clustering baselines. From the tables, we draw the following key observations. *On the one hand*, compared to autoencoder-based methods, contrastive learning-based approaches show better performance. The reason lies in the ability of contrastive learning to more effectively exploit the intrinsic semantic information of the graph-structured data. By learning discriminative representations in a principled manner, contrastive learning better serves the clustering task. *On the other hand*, our approach achieves almost the best results on all five datasets, and

significantly outperforms the runner-ups on many datasets. For instance, on Cora and BAT in Table 1, our proposed CoCo surpass the runner-ups by {4.13%, 1.45%, 2.00%, 5.23%} and {4.16%, 8.74%, 12.79%, 4.19%} under four evaluation metrics, providing substantial evidence for the superiority of our approach. These results substantiate the success of the compactness learning and consistency learning embodied in CoCo for graph clustering, while also implicitly suggesting the superiority of cross-view consistency learning over contrastive learning, which is further validated in detail in Section 3.3 and Appendix H.

### 3.3 Ablation Study

In this section, we analyze the impact of various components of our proposed method.

**Comparison of different model variants.** We first define different model variants as: (i) $M_1$: solely adopt the local Laplacian smoothing filter to extract node representations (*i.e.*, $\tilde{\mathbf{X}}^l$) for clustering; (ii) $M_2$: solely adopt the global one (*i.e.*, $\tilde{\mathbf{X}}^g$) for clustering; (iii) $M_3$: adopt local low-rank embeddings (*i.e.*, $\tilde{\mathbf{Z}}^l$) by compactness learning based on $\tilde{\mathbf{X}}^l$ for clustering; (iv) $M_4$: adopt global low-rank embeddings (*i.e.*, $\tilde{\mathbf{Z}}^g$) by compactness learning based on $\tilde{\mathbf{X}}^g$ for clustering; (v) $M_5$: remove compactness learning from our full model CoCo. The results are summarized in Figure 2.

Comparing $M_3$ with $M_1$ and $M_4$ with $M_2$, mapping raw node features to low-rank embeddings improves the performance in both cases, indicating the effectiveness of our low-rank representations in learning better cluster assignments. Similarly, when comparing $M_5$ with our CoCo, removing the low-rank mapping also leads to performance degradation, further emphasizing the necessity of compactness learning. In addition, comparing $M_5$ and CoCo with $M_1$-$M_4$, we observe a significant performance difference between the two groups, possibly because our consistency learning effectively integrates semantic knowledge from both local and global perspectives, enabling more discriminative and robust representations compared to any single viewpoint. Below, we further consider the comparison between the consistency loss and other surrogate losses.

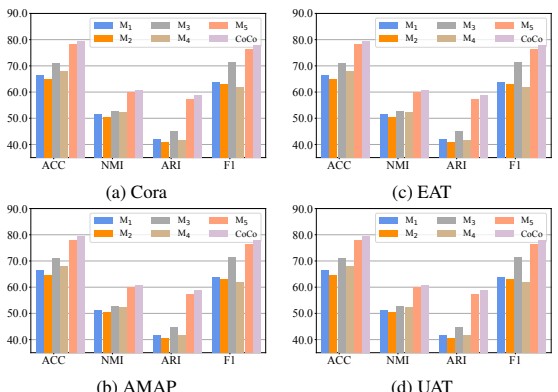

Figure 2: The ablation experimental results.

**Influence of Consistency Learning.** To investigate the advantages of our proposed consistency learning, we compare two widely used loss functions, Mean Squared Error (MSE) and contrastive loss InfoNCE (Chen et al., 2020b), on all datasets. The comparative results are presented in Table 2 and Appendix H. It can be observed that InfoNCE performs worse than MSE in most cases. This could be potentially attributed to InfoNCE's reliance on extensive negative instance sampling in

Table 2: The comparative results of consistency learning loss $v.s$ MSE and InfoNCE.

| Dataset | Loss | ACC | NMI | ARI | F1 |
|---|---|---|---|---|---|
| **Cora** | MSE | 77.84±0.67 | 60.31±0.89 | 57.81±1.42 | 73.89±1.09 |
| | InfoNCE | 75.57±1.16 | 58.03±1.44 | 54.69±1.85 | 72.58±1.81 |
| | Consistency | **79.36±0.69** | **60.71±0.59** | **58.76±1.47** | **77.95±0.72** |
| **AMAP** | MSE | 77.62±0.44 | 67.68±0.76 | 58.51±0.97 | 71.84±0.77 |
| | InfoNCE | 77.25±0.33 | 67.12±0.46 | 58.24±0.57 | 71.89±0.53 |
| | Consistency | **79.27±0.70** | **68.85±1.55** | **60.94±1.51** | **72.36±1.15** |
| **UAT** | MSE | 57.18±0.74 | 28.43±0.62 | 25.65±1.11 | 56.96±0.73 |
| | InfoNCE | 56.72±0.23 | 27.67±0.42 | 25.01±0.45 | 56.39±0.39 |
| | Consistency | **59.68±0.36** | **30.12±0.51** | **29.46±0.47** | **58.03±0.34** |

contrastive learning, which may introduce more complexity and sensitivity to hyper-parameter tuning compared to the direct regression-based optimization of MSE. Moreover, the performance of consistency learning significantly outperforms the other two loss functions, which fully demonstrates the effectiveness of semantic enhancement brought by aligning the similarity distributions of local and global perspectives in consistency learning.

### 3.4 Visualization Analysis

To visually verify the validity of our proposed CoCo, we plot the projected distributions ($t$-SNE) of the learned embeddings on Cora and AMAP (Van der Maaten & Hinton, 2008), compared with six

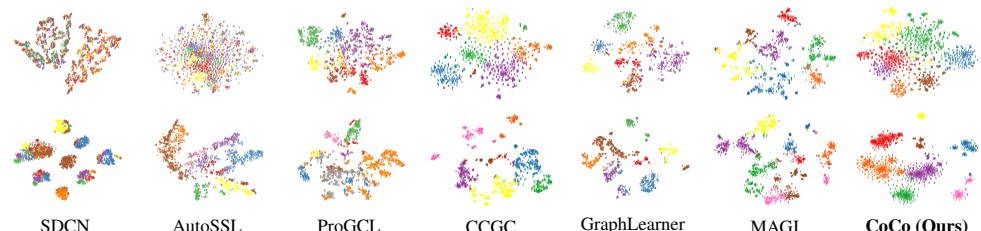

SDCN     AutoSSL     ProGCL     CCGC     GraphLearner     MAGI     **CoCo (Ours)**

Figure 3: The $t$-SNE results comparing our CoCo with competitive baselines on two datasets. The first row and second row correspond to Cora and AMAP, respectively.

baselines. The visualization results are displayed in Figure 3. It can be observed that CoCo exhibits a lower degree of inter-cluster confusion on Cora and better separability among different clusters on AMAP, showcasing CoCo's ability to learn more discriminative representations and effective cluster assignments compared with competitive methods.

## 3.5 SENSITIVITY ANALYSIS

Here, we study the sensitivity of key hyper-parameters: the subspace dimension $K$, the anchor number $M$, and the teleport probability $\alpha$. The results are presented in Figure 4 and Appendix I. We also perform robust analysis with varying $t$ (in Equation 1) in Appendix J.

**Effect of $K$.** As shown in the first row of Figure 4, when $K$ ranges from 32 to 128, the clustering performance under four metrics on two datasets increases slowly. However, as $K$ further increases, the performance of the model starts to decline. This is because incorporating too high-dimensional subspace introduces redundant or irrelevant underlying structural information, which hinders the compactness of node embeddings.

**Effect of $M$.** The second row of Figure 4 reports the impact of the number of anchor samples $M$ in the queue. It can be observed that when $M$ is very small ($M$=64), the model's performance is poorer. This is because for each node, there are not enough anchor samples in its vicinity to adequately represent the node's neighborhood structure. As $M$ increases, the model maintains relatively high performance and remains stable. This allows the model to better capture the neighborhood structure information of nodes and enhances consistency learning.

**Effect of $\alpha$.** In the third row of Figure 4, we report the impact of the teleport probability $\alpha$ in graph diffusion matrix $\mathbf{S}$. It can be observed that the model performance remains relatively stable when $\alpha$ is set to 0.1 or 0.2. However, as $\alpha$ continues to increase, the performance begins to decline, with a particularly noticeable drop at $\alpha$ = 0.8. This is because a larger $\alpha$ leads to a more "localized" diffusion scope (prioritizing the node's own information), causing the two branches to capture increasingly similar information and thus failing to provide additional benefits. In contrast, a smaller $\alpha$ results in a more "global" diffusion (integrating information across the entire graph), which is more critical for capturing richer and complementary knowledge.

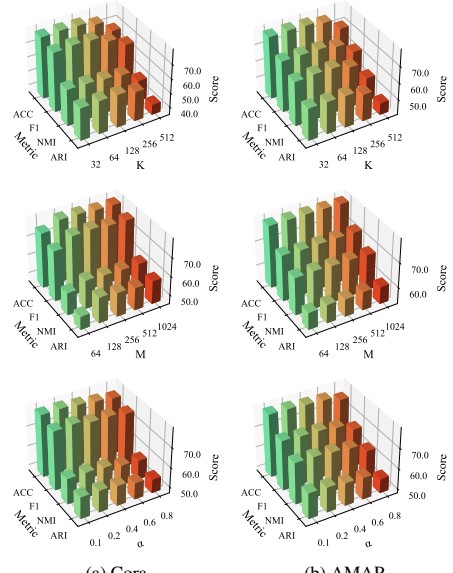

(a) Cora     (b) AMAP

Figure 4: Sensitivity experimental results.

## 3.6 ROBUSTNESS ANALYSIS ON NOISY GRAPHS

To further assess the effectiveness and robustness of CoCo, we construct noisy graphs from BAT under three noise settings: (1) *attribute noise* by adding Gaussian noise $\mathcal{N}(0, 5)$; (2) *edge noise* by

Table 4: Comparison on heterophilic graphs.          Table 5: Comparison on homophilic graphs.

| Dataset | GraphLearner | | MAGI | | CoCo (Ours) | |
|---|---|---|---|---|---|---|
| | ACC | NMI | ACC | NMI | ACC | NMI |
| Cornell | 43.00 | 14.17 | 34.26 | 7.63 | **59.68** | **24.34** |
| Wisconsin | 46.06 | **19.38** | 33.39 | 8.40 | **56.39** | 17.04 |

| Dataset | DGCN | | CoCo (Ours) | |
|---|---|---|---|---|
| | ACC | NMI | ACC | NMI |
| Cora | 72.89 | 56.82 | **79.36** | **60.71** |
| AMAP | 76.06 | 65.36 | **79.27** | **68.85** |

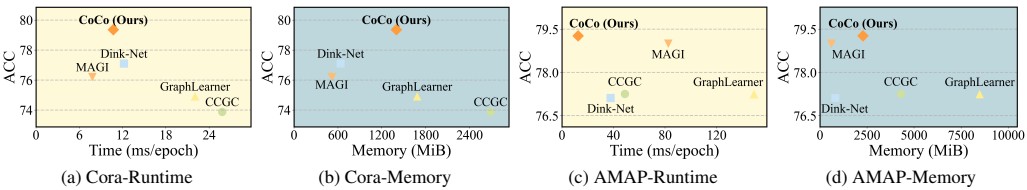

(a) Cora-Runtime  (b) Cora-Memory  (c) AMAP-Runtime  (d) AMAP-Memory

Figure 5: Comparisons of average training time per epoch and memory cost.

randomly adding/deleting edges with probability 0.3; and (3) *both attribute and edge noise*. The clustering accuracy results in Table 3 show that, although noise degrades performance for all methods (cf. Table 1), our CoCo still consistently outperforms GraphLearner and MAGI. It highlights the strong redundancy-reduction and noise-resistant ability of compactness learning, as well as the role of consistency learning in capturing richer semantic information to ensure robustness when facing noise.

Table 3: Comparison of accuracy on noisy BAT.

| Noise Type | Attribute | Edge | Attribute & Edge |
|---|---|---|---|
| GraphLearner | 52.52 | 36.79 | 38.63 |
| MAGI | 36.26 | 38.70 | 35.19 |
| CoCo (Ours) | **64.12** | **55.57** | **48.78** |

### 3.7 PERFORMANCE COMPARISON ON HETEROPHILIC GRAPHS

To further validate the applicability of our method, we compare the proposed CoCo with competitive baselines on the heterophilic graphs Cornell and Wisconsin (Pei et al., 2020), and we also compare CoCo on homophilic graphs with method tailored for homophilic graphs, i.e., DGCN (Pan & Kang, 2023). From Table 4, our CoCo mostly outperforms the recent competitive baselines GraphLearner and MAGI on heterophilic graphs, likely due to its robustness against cross-class edges via compactness and consistency learning. From Table 5, CoCo also surpasses DGCN on homophilic graphs, further confirming the effectiveness and applicability of our CoCo.

### 3.8 TIME AND SPACE COMPLEXITY COMPARISON

In this part, we compare the time and space complexities of our CoCo with several latest and competitive methods, i.e., CCGC, Dink-Net, GraphLearner, MAGI, in Figure 5. The results show a comparison of clustering accuracy (ACC, ↑) and average runtime/memory cost (↓) among different methods. Our CoCo achieves best clustering performance on Cora and AMAP while maintaining relatively low time and space complexity, which further demonstrates the efficiency and scalability of our approach. Detailed theoretical analysis and comparison are provided in Appendix E.

Besides, to explore the impact of EM iterations on efficiency, we here we examine how different iteration counts affect performance (using NMI as an example) and efficiency (EM time per iteration / total time per iteration), with the results presented in Table 6. From the results, we can see that although the proportion of EM iteration time in the total runtime gradually increases with the number of iterations, the performance stabilizes after 10 iterations, indicat-

Table 6: Impact of EM iterations.

| Iterations | Cora | | AMAP | |
|---|---|---|---|---|
| | NMI | Efficiency | NMI | Efficiency |
| 1 | 47.99±4.07 | 1.62% | 62.96±2.16 | 2.23% |
| 5 | 59.81±0.73 | 2.93% | 68.50±1.25 | 3.27% |
| 10 | 60.71±0.59 | 4.28% | 68.85±1.55 | 4.24% |
| 20 | 60.69±0.57 | 6.69% | 68.87±1.88 | 6.16% |
| 30 | 60.70±0.60 | 8.46% | 68.81±1.70 | 8.42% |

ing that the model has already converged. Further increasing the number of EM iterations yields no additional benefits. Therefore, it can be concluded that the EM iterative algorithm accounts for only about 4% of the total training time, demonstrating remarkably high efficiency.

## 3.9 Evaluation on Node Classification Task

We further analyze the generalization ability of our CoCo to verify whether the learned node representations perform well on downstream tasks beyond graph clustering task, taking node classification as an example. We evaluate its performance under the predicted accuracy metric using widely adopted datasets including WikiCS (Mernyei & Cangea, 2020), Computers (McAuley et al., 2015), Photo (McAuley et al., 2015), Coauthor CS (Sinha et al., 2015), and Coauthor Physics (Sinha et al., 2015), and compare it with several compet-

Table 7: Performance comparison on node classification task (OOM denotes Out of Memory).

| Dataset | WikiCS | Computers | Photo | Coauthor CS | Coauthor Physics |
|---------|--------|-----------|-------|-------------|------------------|
| GCN | 77.19±0.12 | 86.51±0.54 | 92.42±0.22 | 93.03±0.31 | 95.65±0.16 |
| node2vec | 71.79±0.05 | 84.39±0.08 | 89.67±0.12 | 85.08±0.03 | 91.19±0.04 |
| DeepWalk | 74.35±0.06 | 85.68±0.06 | 89.44±0.11 | 84.61±0.22 | 91.77±0.15 |
| DGI | 75.35±0.14 | 83.95±0.47 | 91.61±0.22 | 92.15±0.63 | 94.51±0.52 |
| GMI | 74.85±0.08 | 82.21±0.31 | 90.68±0.17 | OOM | OOM |
| MVGRL | 77.52±0.08 | 87.52±0.11 | 91.74±0.07 | 92.11±0.12 | 95.33±0.03 |
| GCA | 78.30±0.62 | 88.49±0.51 | 92.99±0.27 | 92.76±0.16 | OOM |
| GRACE | 78.25±0.65 | 88.15±0.43 | 92.52±0.32 | 92.60±0.11 | OOM |
| CCA-SSG | 77.88±0.41 | 87.01±0.41 | 92.59±0.25 | 92.77±0.17 | 95.16±0.10 |
| BGRL | 79.36±0.53 | 88.35±0.32 | 92.87±0.27 | 91.72±0.21 | 95.43±0.09 |
| GTCA | 79.58±0.65 | 89.15±0.57 | 92.97±0.31 | 92.33±0.18 | 95.24±0.07 |
| CoCo | 80.06±0.45 | 89.62±0.43 | 93.34±0.37 | 93.07±0.13 | 95.52±0.08 |

itive state-of-the-art methods, i.e., GCN (Kipf & Welling, 2017), node2vec (Grover & Leskovec, 2016), DeepWalk (Perozzi et al., 2014), DGI (Velickovic et al., 2019), GMI (Peng et al., 2020), MVGRL (Hassani & Khasahmadi, 2020), GCA (Zhu et al., 2020), GRACE (Zhu et al., 2021), CCA-SSG (Zhang et al., 2021), BGRL (Thakoor et al., 2022) and GTCA (Liang et al., 2025a). Specifically, we first perform unsupervised pre-training using the proposed framework, followed by supervised fine-tuning on a labeled dataset. As shown in Table 7, our proposed CoCo outperforms all baseline methods across all datasets, demonstrating the strong generalizability of our learned node representations. This superior performance can be attributed to our method's ability to capture structural information from multiple perspectives and learn low-rank node representations that align more closely with the data distribution, effectively supporting various downstream tasks.

## 4 Conclusion

In this paper, we propose a novel approach named CoCo for deep graph clustering. CoCo first encodes the attribute and topology information from local and global views. Then CoCo exploits low-rank embeddings via GMM to remove noise and redundancy, thereby uncovering the intrinsic structure of nodes. Based on the compact low-rank embeddings, CoCo also performs consistency learning of node similarities to enrich semantics information. Experiments under different data types (homophilic, heterophilic, and noisy graphs) and different tasks (graph clustering mainly and node classificaion) show that our CoCo outperforms the state-of-the-art methods. In addition, the comparison of spatiotemporal complexity demonstrates the efficiency of our method. For future work, we aim to extend our model to temporal graph clustering and single-cell genomics clustering, facilitating the analysis of dynamic structures and accurate grouping of cells by genetic profiles.

## Acknowledgments

This work is supported in part by the National Natural Science Foundation of China under Grants 62306014, 12501344 and 12131001, the Fundamental Research Funds for the Central Universities, LPMC, and KLMDASR, the Postdoctoral Fellowship Program (Grade A) of CPSF under Grant BX20250376 and BX20240239, the China Postdoctoral Science Foundation under Grant 2024M762201, the Sichuan Science and Technology Program under Grant 2025ZNSFSC1506 and 2025ZNSFSC0808, and the Sichuan University Interdisciplinary Innovation Fund.

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

# A EXPECTATION-MAXIMIZATION ALGORITHM FOR GAUSSIAN MIXTURE MODEL

The Gaussian mixture model (GMM) (Richardson & Green, 1997) is a linear superposition of multiple Gaussian components. Assuming GMM consists of $K$ Gaussians, given the observed data $\mathbf{X} = (\mathbf{x}_1, \ldots, \mathbf{x}_N)^\top$, the likelihood function is given by

$$p(\mathbf{X}|\boldsymbol{\theta}) = \prod_{i=1}^{N} \left( \sum_{k=1}^{K} \pi_k \mathcal{N}(\mathbf{x}_i|\boldsymbol{\mu}_k, \boldsymbol{\Sigma}_k) \right),$$

where $\boldsymbol{\theta} = \{\pi_k, \boldsymbol{\mu}_k, \boldsymbol{\Sigma}_k\}_{k=1}^{K}$, $\mathcal{N}(\mathbf{x}|\boldsymbol{\mu}_k, \boldsymbol{\Sigma}_k)$ is the probability density function of the $k$-th Gaussian, and $\pi_k$ is the prior probability. We can employ the expectation-maximization (EM) algorithm (Dempster et al., 1977) to fit GMM, in which the latent variables $\mathbf{Y} = (y_{ik}) \in \mathbb{R}^{N \times K}$ are introduced to indicate whether the $i$-th sample belongs to the $k$-th Gaussian by $y_{ik}$.

In the E step, EM computes the posterior probability:

$$\gamma(y_{ik}) = p(y_{ik} = 1 | \mathbf{x}_i, \boldsymbol{\theta}^{\text{old}}) = \frac{\pi_k^{\text{old}} \mathcal{N}(\mathbf{x}_i | \boldsymbol{\mu}_k^{\text{old}}, \boldsymbol{\Sigma}_k^{\text{old}})}{\sum_{j=1}^{K} \pi_j^{\text{old}} \mathcal{N}(\mathbf{x}_i | \boldsymbol{\mu}_j^{\text{old}}, \boldsymbol{\Sigma}_j^{\text{old}})},$$

and the posterior expectation:

$$Q(\boldsymbol{\theta}, \boldsymbol{\theta}^{\text{old}}) = \sum_{k=1}^{K} \sum_{i=1}^{N} \gamma(y_{ik})(\log \pi_k + \log \mathcal{N}(\mathbf{x}_i | \boldsymbol{\mu}_k, \boldsymbol{\Sigma}_k)).$$

In the M step, EM maximizes $Q(\boldsymbol{\theta}, \boldsymbol{\theta}^{\text{old}})$ to achieve the updated parameters $\boldsymbol{\theta}^{\text{new}}$:

$$\boldsymbol{\mu}_k^{\text{new}} = \frac{1}{\sum_{i=1}^{N} \gamma(y_{ik})} \sum_{i=1}^{N} \gamma(y_{ik}) \mathbf{x}_i, \quad \pi_k^{\text{new}} = \frac{\sum_{i=1}^{N} \gamma(y_{ik})}{N},$$

$$\boldsymbol{\Sigma}_k^{\text{new}} = \frac{1}{\sum_{i=1}^{N} \gamma(y_{ik})} \sum_{i=1}^{N} \gamma(y_{ik})(\mathbf{x}_i - \boldsymbol{\mu}_k^{\text{new}})(\mathbf{x}_i - \boldsymbol{\mu}_k^{\text{new}})^\top.$$

# B PROOFS

## B.1 PROOF OF THEOREM 1.

*Proof.* The smoothness of a graph signal $\mathbf{x}$, defined on the vertices of the graph, can be characterized by the Rayleigh quotient based on the normalized graph Laplacian matrix $\tilde{\mathbf{L}}$ (Horn & Johnson, 2012), i.e., $\mathbf{x}^\top \tilde{\mathbf{L}} \mathbf{x} / \mathbf{x}^\top \mathbf{x}$, which expresses the normalized variance score of $\mathbf{x}$. The smaller this value is, the smoother the signal is. Denote by $\tilde{\mathbf{U}}^{\text{p}} = (\tilde{\mathbf{u}}_1^{\text{p}}, \ldots, \tilde{\mathbf{u}}_N^{\text{p}})^\top$ and $\tilde{\boldsymbol{\Lambda}}^{\text{p}} = \text{diag}(\tilde{\lambda}_1^{\text{p}}, \ldots, \tilde{\lambda}_N^{\text{p}})$ the eigenvector matrix and eigenvalue matrix of $\tilde{\mathbf{L}}^{\text{p}}(\text{p} = \text{l}, \text{g})$. Based on the eigendecomposition of $\tilde{\mathbf{L}}^{\text{p}}$, the signal $\mathbf{x}$ can be decomposed into:

$$\tilde{\mathbf{U}}^{\text{p}} \mathbf{c}^{\text{p}} = \sum_{i=1}^{N} c_i^{\text{p}} \tilde{\mathbf{u}}_i^{\text{p}} \text{ (Fourier inverse transform)},$$

where $\mathbf{c}^{\text{p}}$ is the coefficient vector. Then the filtered signal by Equation 1 is $\tilde{\mathbf{x}}^{\text{p}} = \sum_{i=1}^{N} (1 - \tilde{\lambda}_i^{\text{p}}/k^{\text{p}})^t c_i^{\text{p}} \tilde{\mathbf{u}}_i^{\text{p}}$ and the corresponding Rayleigh quotient is:

$$\frac{\tilde{\mathbf{x}}^{\text{p}\top} \tilde{\mathbf{L}}^{\text{p}} \tilde{\mathbf{x}}^{\text{p}}}{\tilde{\mathbf{x}}^{\text{p}\top} \tilde{\mathbf{x}}^{\text{p}}} = \frac{\sum_{i=1}^{N} [(1 - \tilde{\lambda}_i^{\text{p}}/k^{\text{p}})^t c_i^{\text{p}}]^2 \tilde{\lambda}_i^{\text{p}}}{\sum_{i=1}^{N} [(1 - \tilde{\lambda}_i^{\text{p}}/k^{\text{p}})^t c_i^{\text{p}}]^2}.$$

To filter high-frequency noise, $k^{\text{p}}$ should be set to ensure the low-pass property of the filter (for any $t$), also the smoothness. On the one hand, $1 - \tilde{\lambda}_i^{\text{p}}/k^{\text{p}}$ should be non-negative. Hence, $k^{\text{p}} \geq \tilde{\lambda}_i^{\text{p}}$ for any $1 \leq i \leq N$, which leads to $k^{\text{p}} \geq \tilde{\lambda}_{\max}^{\text{p}}$. On the other hand, $1 - \tilde{\lambda}_i^{\text{p}}/k^{\text{p}}$ should become smaller as $\tilde{\lambda}_i^{\text{p}}$ becomes larger to ensure that the filter captures low-frequency signal. So the value of $k^{\text{p}}$ should be small and the optimal $k^{\text{p}}$ is set to $\tilde{\lambda}_{\max}^{\text{p}}$, which completes the proof. $\square$

### B.2 PROOF OF REMARK 1.

*Proof.* By the conditional probability formula, we have:

$$\log p(\mathbf{Z}|\boldsymbol{\lambda}^{\text{new}}) = \log p(\mathbf{Z}, \mathbf{Y}|\boldsymbol{\lambda}^{\text{new}}) - \log p(\mathbf{Y}|\mathbf{Z}, \boldsymbol{\lambda}^{\text{new}}).$$

Hence, we further have:

$$\log p(\mathbf{Z}|\boldsymbol{\lambda}^{\text{new}}) = \int p(\mathbf{Y}|\mathbf{Z}, \boldsymbol{\lambda}^{\text{old}}) \log p(\mathbf{Z}|\boldsymbol{\lambda}^{\text{new}}) d\mathbf{Y}$$

$$= \int p(\mathbf{Y}|\mathbf{Z}, \boldsymbol{\lambda}^{\text{old}}) \log p(\mathbf{Z}, \mathbf{Y}|\boldsymbol{\lambda}^{\text{new}}) d\mathbf{Y} -$$

$$\int p(\mathbf{Y}|\mathbf{Z}, \boldsymbol{\lambda}^{\text{old}}) \log p(\mathbf{Y}|\mathbf{Z}, \boldsymbol{\lambda}^{\text{new}}) d\mathbf{Y}$$

$$\triangleq Q(\boldsymbol{\lambda}^{\text{new}}, \boldsymbol{\lambda}^{\text{old}}) - H(\boldsymbol{\lambda}^{\text{new}}, \boldsymbol{\lambda}^{\text{old}}).$$

If $Q(\boldsymbol{\lambda}^{\text{new}}, \boldsymbol{\lambda}^{\text{old}}) \geq Q(\boldsymbol{\lambda}^{\text{old}}, \boldsymbol{\lambda}^{\text{old}})$ and $H(\boldsymbol{\lambda}^{\text{new}}, \boldsymbol{\lambda}^{\text{old}}) \leq H(\boldsymbol{\lambda}^{\text{old}}, \boldsymbol{\lambda}^{\text{old}})$ hold, we can conclude that:

$$\log p(\mathbf{Z}|\boldsymbol{\lambda}^{\text{new}}) \geq \log p(\mathbf{Z}|\boldsymbol{\lambda}^{\text{old}}).$$

In the E step, we maximized the $Q$-function, which implies that $Q(\boldsymbol{\lambda}^{\text{new}}, \boldsymbol{\lambda}^{\text{old}}) \geq Q(\boldsymbol{\lambda}^{\text{old}}, \boldsymbol{\lambda}^{\text{old}})$. As for the H-function, we have:

$$H(\boldsymbol{\lambda}^{\text{new}}, \boldsymbol{\lambda}^{\text{old}}) - H(\boldsymbol{\lambda}^{\text{old}}, \boldsymbol{\lambda}^{\text{old}})$$

$$= \int p(\mathbf{Y}|\mathbf{Z}, \boldsymbol{\lambda}^{\text{old}}) \log \frac{p(\mathbf{Y}|\mathbf{Z}, \boldsymbol{\lambda}^{\text{new}})}{p(\mathbf{Y}|\mathbf{Z}, \boldsymbol{\lambda}^{\text{old}})} d\mathbf{Y}$$

$$\leq \log \int p(\mathbf{Y}|\mathbf{Z}, \boldsymbol{\lambda}^{\text{new}}) d\mathbf{Y} = \log 1 = 0,$$

which follows from the Jensen inequality, i.e., $E(\log X) \leq \log E(X)$. The proof is completed. $\square$

### B.3 PROOF OF THEOREM 2.

*Proof.* After the subspace training algorithm converges, according to Equation 3 and summing over $k$, we have:

$$\sum_{k=1}^{K} \hat{\lambda}_{ik} \sum_{j=1}^{D} \hat{\gamma}(y_{jk}) = \sum_{k=1}^{K} \sum_{j=1}^{D} \hat{\gamma}(y_{jk}) z_{ij}.$$

Then the following equality also holds:

$$\sum_{j=1}^{D} \left( \sum_{k=1}^{K} \hat{\lambda}_{ik} \hat{\gamma}(y_{jk}) \right) = \sum_{j=1}^{D} \left( \sum_{k=1}^{K} \hat{\gamma}(y_{jk}) \right) z_{ij}.$$

It implies that $\sum_{j=1}^{D} \hat{z}_{ij} = \sum_{j=1}^{D} z_{ij}$, which follows from that $\sum_{k=1}^{K} \hat{\gamma}(y_{jk}) = 1$ and Equation 4 holds. This guarantees that the total mass of each row in $\mathbf{Z}$ is preserved, meaning that the aggregated information associated with individual $i$ remains unchanged after reconstruction. Also, it ensures that the individual-level signal encoded in the original embedding is retained.

In addition, the Q-function defined in the proof of Remark 1 can be rewritten as:

$$Q(\boldsymbol{\lambda}, \boldsymbol{\lambda}^{\text{old}}) = \sum_{j=1}^{D} \sum_{k=1}^{K} \gamma(y_{jk}) \log \left[ \frac{1}{(2\pi)^{\frac{\bar{N}}{2}} \sigma^{\frac{1}{2}}} \exp\left( -\frac{||\mathbf{z}_{\cdot,j} - \boldsymbol{\lambda}_{\cdot,k}||^2}{2\sigma} \right) \right]$$

$$= \sum_{j=1}^{D} \sum_{k=1}^{K} \gamma(y_{jk}) \log \frac{1}{(2\pi)^{\frac{\bar{N}}{2}} \sigma^{\frac{1}{2}}} + \sum_{j=1}^{D} \sum_{k=1}^{K} \gamma(y_{jk}) \cdot \frac{||\mathbf{z}_{\cdot,j} - \boldsymbol{\lambda}_{\cdot,k}||^2}{2\sigma}$$

$$= -D \log[(2\pi)^{\frac{\bar{N}}{2}} \sigma^{\frac{1}{2}}] + \frac{1}{2\sigma} \sum_{i=1}^{\bar{N}} \sum_{j=1}^{D} \sum_{k=1}^{K} p(y_{jk} = 1 | \mathbf{z}_{\cdot,j}, \boldsymbol{\lambda}_{\cdot,k})(z_{ij} - \lambda_{ik})^2,$$

which is maximized in the M step. Obviously, continuously maximizing Q-function throughout the entire iteration process is equivalent to minimizing the objective:

$$\sum_{i=1}^{\bar{N}}\sum_{j=1}^{D}\sum_{k=1}^{K} p(y_{jk} = 1 | \mathbf{z}_{\cdot,j}, \boldsymbol{\lambda}_{\cdot,k})(z_{ij} - \lambda_{ik})^2. \tag{8}$$

Again, when converging, due to that $\sum_{k=1}^{K} \hat{\gamma}(y_{jk}) = 1$ and Equation 4 holds, the optimized objective in Equation 8 can be written as:

$$\sum_{i=1}^{\bar{N}}\sum_{j=1}^{D}\left( z_{ij}^2 + \sum_{k=1}^{K}\hat{\gamma}(y_{jk})\hat{\lambda}_{ik}^2 - 2z_{ij}\hat{z}_{ij}\right).$$

As for the term $\sum_{k=1}^{K}\hat{\gamma}(y_{jk})\hat{\lambda}_{ik}^2$, we have:

$$\sum_{j=1}^{D}\sum_{k=1}^{K}\hat{\gamma}(y_{jk})\hat{\lambda}_{ik}^2 = \sum_{k=1}^{K}\hat{\lambda}_{ik}^2\sum_{j=1}^{D}\hat{\gamma}(y_{jk})$$

$$= \sum_{k=1}^{K}\hat{\lambda}_{ik}^2 \cdot \frac{\sum_{j=1}^{D}\hat{\gamma}(y_{jk})z_{ij}}{\hat{\lambda}_{ik}} \qquad \text{(by Equation 3)}$$

$$= \sum_{j=1}^{D} z_{ij}\sum_{k=1}^{K}\hat{\lambda}_{ik}\hat{\gamma}(y_{jk}) = \sum_{j=1}^{D} z_{ij}\hat{z}_{ij}.$$

Thus, we can obtain that the objective in Equation 8 is minimized to

$$\sum_{i=1}^{\bar{N}}\sum_{j=1}^{D} z_{ij}(z_{ij} - \hat{z}_{ij}) = \underbrace{\sum_{i,j} z_{ij}^2}_{\text{original total variation}} - \underbrace{\sum_{i,j} z_{ij}\hat{z}_{ij}}_{\text{alignment with reconstruction}}.$$

The first term, $\sum_{i,j} z_{ij}^2$, measures the total variation (energy) of the original embedding, while $\sum_{i,j} z_{ij}\hat{z}_{ij}$ quantifies how much of this variation is preserved in the reconstructed representation. Notably, compared with the self-energy $\sum_{i,j}(\hat{z}_{ij})^2$, the cross-term $z_{ij}\hat{z}_{ij}$ directly captures how well the reconstructed embedding aligns with the original signal rather than merely measuring its own magnitude. Hence, maximizing the cross-correlation term corresponds to keeping the dominant variations present in $\mathbf{Z}$, and the above minimized objective brings a reconstruction $\hat{\mathbf{Z}}$ that preserves as much of the original total variation as possible while suppressing redundant or noisy components, which completes the proof. □

## C MORE DISCUSSIONS ON THE PROPOSED COCO

**GMM *v.s.* Other optional Rank Reduction Methods.** In order to obtain low-rank representations, many methods can achieve this goal. However, the advantage of GMM lies in its linear modeling complexity, with time and space complexities of $O(NDK)$ and $O(ND)$, respectively, while PCA has time and space complexities of $O(ND^2 + D^3)$ and $O(ND + D^2)$. When the number of features is high, the complexity of PCA is also high. Compared to K-means, since GMM is a probabilistic model and belongs to soft clustering algorithms, it is more robust to outliers and can be applied to a wider range of data distributions. Compared to methods like autoencoders for dimensionality reduction, GMM has fewer training parameters and lower time and space complexities. Therefore, our paper chooses GMM to achieve low-rank representation learning. Of course, other better ways to conduct low-rank space training can be further explored.

**Consistency Learning *v.s.* Contrastive Learning.** Contrastive learning (Chen et al., 2020b; You et al., 2020) shares similar ideas with our consistency learning, utilizing the similarity of data representations under different data augmentations to bring similar samples closer and push dissimilar samples apart. However, there are significant differences between contrastive learning and our consistency learning. First, the proposed consistency learning aims to uncover relationship information

among non-iid nodes, thereby better reflecting the underlying structural distribution of nodes in the representation. Second, we consider consistency learning at the relational information level from both local and global perspectives, emphasizing alignment learning. This enables representations from two perspectives to interact, guide each other, and learn, thereby obtaining more semantically rich node representations to better serve graph clustering. Third, contrastive learning involves negative samples, whereas our consistency learning does not. In summary, contrastive learning focuses on sample discriminability at the instance level, whereas our consistency learning considers to bridge the gap between the considered local and global views at the relational distribution level. This is a higher-order, more global approach to capturing semantic knowledge in graph data.

Besides, in Table 2 of the experiments section and Appendix H, we also compare the performance of our consistency learning loss with the contrastive learning loss (InfoNCE) (Chen et al., 2020b; You et al., 2020). It's clear that our consistency learning outperforms contrastive learning by a considerable margin, which fully demonstrates that our distribution-based consistency is more capable of capturing the intrinsic dependencies between nodes at a higher semantic level.

## D TRAINING PROCEDURE

The training procedure of the whole framework is shown in Algorithm 1 below.

---

**Algorithm 1** The Optimization Framework of CoCo

---

**Input**: Graph $\mathcal{G} = \{\mathcal{V}, \mathcal{E}, \mathbf{X}, \mathbf{A}\}$; Cluster number $C$;
**Output**: Clustering result $\mathbf{c}$;

1: Obtain the local- and global-view filtered attribute matrix $\tilde{\mathbf{X}}^l$ and $\tilde{\mathbf{X}}^g$ in Equation 1;
2: Initialize the trainable parameters in $\text{MLP}_1$ and $\text{MLP}_2$;
3: **while** not converge **do**
4:     Update $\mathbf{Z}^l$ and $\mathbf{Z}^g$ by encoding $\tilde{\mathbf{X}}^l$ and $\tilde{\mathbf{X}}^g$ in the unshared MLP encoders;
5:     Update the optimal subspace $\hat{\mathbf{\Lambda}}$ by performing iterations through Equation 3;
6:     Update the reconstructed embeddings $\tilde{\mathbf{Z}}^l$ and $\tilde{\mathbf{Z}}^g$ by Equation 4 and Equation 5;
7:     Update the similarity distributions $\mathbf{p}^i$ and $\mathbf{q}^i$ for $i = 1, \dots, N$;
8:     Calculate the loss $\mathcal{L}$ in Equation 6;
9:     Perform backpropagation and update the entire network in CoCo by minimizing $\mathcal{L}$;
10: **end while**
11: Derive clustering result $\mathbf{c}$ by applying $K$-means to the fused representations from Equation 7;

---

## E COMPLEXITY ANALYSIS

Given a graph dataset comprising $N$ nodes and $E$ edges. The sparse diffusion matrix $\hat{S}$ can be approximately calculated in linear time and space, i.e., $O(N)$ (Gasteiger et al., 2019). Assume the number of non-zero elements in the sparse diffusion matrix $\hat{S}$ is denoted as $nnz(\hat{S})$, the computational complexity of graph convolutional filters is $O(nnz(\hat{S}) * N)$. For

Table 8: Theoretical analysis of complexity

| Method | Time | Space |
|---|---|---|
| MVGRL | $O(N^2D)$ | $O(N^2)$ |
| RGC | $O(N^2D)$ | $O(N^2)$ |
| CCGC | $O(N^2D)$ | $O(N^2)$ |
| Dink-Net | $O(NCD)$ | $O(NC + ND)$ |
| GraphLearner | $O(N^2D)$ | $O(N^2)$ |
| MAGI | $O(ND^2)$ | $O(ND + |\mathcal{E}| + N^2)$ |
| **CoCo (Ours)** | $O(NDK + NMD)$ | $O(ND + NM)$ |

each dataset, they only need to be computed once at the beginning of training. In each epoch, GMM training and feature reconstruction have linear time complexity of $O(NDK)$ and space complexity of $O(ND)$. Consistency learning has time complexity of $O(NMD)$ and space complexity of $O(NM)$. In summary, the preprocessing time and space complexities of CoCo based on sparsification are $O(N)$ and $O(E)$, respectively. In each training epoch, the time complexity of CoCo is $O(NDK + NMD)$ and the space complexity of CoCo is $O(ND + NM)$.

Comparison between our CoCo and several recent and competitive baselines is presented in the Table 8. It can be observed that, comparing with the quadratic time complexity of MVGRL, RGC,

CCGC and GraphLearner, our CoCo scales linearly with the number of samples, making it more suitable for large-scale datasets where computational efficiency is crucial.

## F  DETAILS OF EXPERIMENTAL SETUP

**Datasets.** Following Yang et al. (2023), we assess the performance of our CoCo with five widely used benchmark datasets for deep graph clustering, i.e., Cora (Sen et al., 2008), AMAP (Shchur et al., 2018), BAT (Liu et al., 2023c), EAT (Liu et al., 2023c), and UAT (Liu et al., 2023c).

**Baseline Methods.**  To comprehensively assess the effectiveness of our proposed CoCo, we benchmark it against leading state-of-the-art methods, including autoencoder-based deep graph clustering/self-supervised learning methods: DEC (Xie et al., 2016), IDEC (Guo et al., 2017), DAEGC (Wang et al., 2019), ARGA (Pan et al., 2019), SDCN (Bo et al., 2020), DFCN (Tu et al., 2021); and contrastive deep graph clustering/self-supervised learning methods: AGE (Cui et al., 2020), MVGRL (Hassani & Khasahmadi, 2020), GDCL (Zhao et al., 2021), AutoSSL (Jin et al., 2022b), AGC-DRR (Gong et al., 2022), AFGRL (Lee et al., 2022), GDCL (Zhao et al., 2021), ProGCL (Xia et al., 2022), RGC (Liu et al., 2023a), Dink-Net (Liu et al., 2023b), CCGC (Yang et al., 2023), GraphLearner (Yang et al., 2024), and MAGI (Liu et al., 2024a).

**Evaluation Metrics and Implementation Details.** We adopt four benchmark metrics following Bo et al. (2020) for evaluation: Accuracy (ACC), Normalized Mutual Information (NMI), Average Rand Index (ARI), and Macro F1-score (F1). Larger values imply better clustering results. For each method, we present the mean and standard deviation of the four metrics across 10 runs. The proposed model is implemented with PyTorch and all experiments are carried out on NVIDIA GeForce RTX 4090. The hidden dimension $d$ is set to 500 for AMAP/UAT, and 1500 for other datasets. We set the total number of training epochs to 1000 and the number of filter layers to 3. The dimension of the subspace $K$ is set to 64 and the number of the selected anchor samples $M$ is set to 128. The parameters of temperature $\tau$ and teleport probability $\alpha$ are set to 0.02 and 0.2, respectively.

## G  ADDITIONAL MAIN EXPERIMENTAL RESULTS

Table 9: Clustering performance on five benchmark datasets (mean ± standard deviation). The top two results for each method are marked in **bold** and underline, respectively.

| Dataset | Metric | DEC | IDEC | DAEGC | ARGA | AGE | MVGRL | GDCL | AGC-DRR | RGC | Dink-Net | CoCo (Ours) |
|---|---|---|---|---|---|---|---|---|---|---|---|---|
| Cora | ACC | 46.50±0.26 | 51.61±1.02 | 70.43±0.36 | 71.04±0.25 | 73.50±1.83 | 70.47±3.70 | 70.83±0.47 | 40.62±0.55 | - | 77.11±0.10 | **79.36±0.69** |
|  | NMI | 23.54±0.34 | 26.31±1.22 | 52.89±0.69 | 51.06±0.52 | 57.58±1.42 | 55.57±1.54 | 56.30±0.36 | 18.74±0.73 | 57.60+1.36 | 59.76±0.10 | **60.71±0.59** |
|  | ARI | 15.13±0.42 | 22.07±1.53 | 49.63±0.43 | 47.71±0.33 | 48.05±0.72 | 48.70±3.94 | 50.10±2.14 | 14.80±1.64 | 49.46+2.72 | 38.16±0.13 | **58.76±1.47** |
|  | F1 | 39.23±0.17 | 47.17±1.12 | 68.27±0.57 | 69.27±0.39 | 69.28±1.59 | 67.15±1.86 | 52.88±0.97 | 31.23±0.57 | - | 74.76±0.10 | **77.95±0.72** |
| AMAP | ACC | 47.22±0.08 | 47.62±0.08 | 75.96±0.23 | 69.28±2.30 | 75.98±0.68 | 41.07±3.12 | 43.75±0.78 | 76.81±1.45 | - | 79.11±0.43 | **79.27±0.70** |
|  | NMI | 37.35±0.05 | 37.83±0.08 | 65.25±0.45 | 58.36±2.76 | 65.38±0.61 | 30.28±3.94 | 37.32±0.28 | 66.54±1.24 | **69.61+0.36** | 67.70±0.30 | 68.85±1.55 |
|  | ARI | 18.59±0.04 | 19.24±0.07 | 58.12±0.24 | 44.18±4.41 | 55.89±1.34 | 18.77±2.34 | 21.57±0.51 | 60.15±1.56 | 59.58+0.39 | 59.76±0.76 | **60.94±1.51** |
|  | F1 | 46.71±0.12 | 47.20±0.11 | 69.87±0.54 | 64.30±1.95 | 71.74±0.93 | 32.88±5.50 | 38.37±0.29 | 71.03±0.64 | - | **72.86±0.32** | 72.36±1.15 |
| BAT | ACC | 42.09±2.21 | 39.62±0.87 | 52.67±0.00 | 67.86±0.80 | 56.68±0.76 | 37.56±0.32 | 45.42±0.54 | 47.79±0.02 | - | 54.20±0.13 | **78.85±0.91** |
|  | NMI | 14.10±1.99 | 12.80±1.74 | 21.43±0.35 | 49.09±0.54 | 36.04±1.54 | 29.33±0.70 | 31.70±0.42 | 19.91±0.24 | 51.58+0.83 | 28.33±0.32 | **55.00±0.87** |
|  | ARI | 07.99±1.21 | 07.85±1.31 | 18.18±0.29 | 42.02±1.21 | 26.59±1.83 | 13.45±0.03 | 19.33±0.57 | 14.59±0.13 | 47.16+1.35 | 23.00±0.54 | **53.52±1.15** |
|  | F1 | 42.63±2.35 | 40.11±0.99 | 52.23±0.03 | 67.02±1.15 | 55.07±0.80 | 29.64±0.49 | 39.94±0.57 | 42.33±0.51 | - | 53.22±0.17 | **78.56±1.01** |
| EAT | ACC | 36.47±1.60 | 35.56±1.34 | 36.89±0.15 | 52.13±0.00 | 47.26±0.32 | 32.88±0.71 | 33.46±0.18 | 37.37±0.11 | - | 50.78±0.17 | **58.87±0.49** |
|  | NMI | 04.96±1.74 | 04.63±0.97 | 05.57±0.06 | 22.48±1.21 | 23.74±0.90 | 11.72±1.08 | 13.22±0.33 | 07.00±0.85 | **37.77+0.13** | 21.66±0.36 | 34.10±1.26 |
|  | ARI | 03.60±1.87 | 03.19±0.76 | 05.03±0.08 | 17.29±0.50 | 16.57±0.46 | 04.68±1.30 | 04.31±0.29 | 04.88±0.91 | **30.16+0.15** | 18.87±0.66 | 27.91±1.52 |
|  | F1 | 34.84±1.28 | 35.52±1.50 | 34.72±0.16 | 52.75±0.07 | 45.54±0.40 | 25.35±0.75 | 25.02±0.21 | 35.20±0.17 | - | 48.28±0.19 | **58.06±2.64** |
| UAT | ACC | 45.61±1.84 | 46.90±0.17 | 52.29±0.49 | 49.31±0.15 | 52.37±0.42 | 44.16±1.38 | 48.70±0.06 | 42.64±0.31 | - | 57.65±0.09 | **59.68±0.36** |
|  | NMI | 16.63±2.39 | 17.84±0.35 | 21.33±0.44 | 25.44±0.31 | 23.64±0.66 | 21.53±0.94 | 25.10±0.01 | 11.15±0.24 | 28.79±0.35 | 25.28±0.15 | **30.12±0.51** |
|  | ARI | 13.14±1.97 | 16.34±0.40 | 20.50±0.51 | 16.57±0.31 | 20.39±0.70 | 17.12±1.46 | 21.76±0.01 | 09.50±0.25 | 19.89±1.30 | 26.42±0.14 | **29.46±0.47** |
|  | F1 | 44.22±1.51 | 46.51±0.17 | 50.33±0.64 | 50.26±0.16 | 50.15±0.73 | 39.44±2.19 | 45.69±0.08 | 35.18±0.32 | - | 54.52±0.10 | **58.03±0.34** |

Apart from the comparisons with a range of competitive baselines in Table 1, we carry out more extensive comparative experiments herein. Table 9 presents a performance comparison between our proposed CoCo and several other graph clustering or unsupervised learning methods. It can be seen that RGC (Liu et al., 2023a) outperforms our model on certain metrics in AMAP and EAT, possibly because RGC can automatically learn the number of clusters, which may, to some extent, better approximate the true data distribution. However, we still achieve the best performance on most datasets. It can be attributed to the fact that our proposed CoCo captures complementary semantic information from both local and global views, enabling a more comprehensive exploration

of the graph's characteristics. Moreover, our CoCo learns low-rank embeddings, which effectively removes redundancies and noise. Finally, consistency learning enriches the semantics of node representations from both local and global perspectives. Hence, different modules complement each other to facilitate better clustering assignments.

## H  ADDITIONAL RESULTS ON INFLUENCE OF CONSISTENCY LEARNING

Table 10: The comparative analysis of consistency learning across all datasets.

| Dataset | Loss | ACC | NMI | ARI | F1 |
|---------|------|-----|-----|-----|----|
| **Cora** | MSE | 77.84±0.67 | 60.31±0.89 | 57.81±1.42 | 73.89±1.09 |
| | InfoNCE | 75.57±1.16 | 58.03±1.44 | 54.69±1.85 | 72.58±1.81 |
| | Consistency | **79.36+0.69** | **60.71±0.59** | **58.76±1.47** | **77.95±0.72** |
| **AMAP** | MSE | 77.62±0.44 | 67.68±0.76 | 58.51±0.97 | 71.84±0.77 |
| | InfoNCE | 77.25±0.33 | 67.12±0.46 | 58.24±0.57 | 71.89±0.53 |
| | Consistency | **79.27±0.70** | **68.85±1.55** | **60.94±1.51** | **72.36±1.15** |
| **BAT** | MSE | 76.87±0.60 | 53.30±1.18 | 50.85±1.07 | 76.32±0.71 |
| | InfoNCE | 78.70±0.53 | 53.99±0.74 | 51.85±1.14 | 78.03±0.51 |
| | Consistency | **78.85±0.91** | **55.00±0.87** | **53.52±1.15** | **78.56±1.01** |
| **EAT** | MSE | 57.42±0.45 | 32.80±0.86 | 26.27±0.74 | 57.83±0.38 |
| | InfoNCE | 55.26±1.25 | 30.34±0.96 | 23.76±0.94 | 55.53±0.94 |
| | Consistency | **58.87±0.49** | **34.10±1.26** | **27.91±1.52** | **58.06±2.64** |
| **UAT** | MSE | 57.18±0.74 | 28.43±0.62 | 25.65±1.11 | 56.96±0.73 |
| | InfoNCE | 56.72±0.23 | 27.67±0.42 | 25.01±0.45 | 56.39±0.39 |
| | Consistency | **59.68±0.36** | **30.12±0.51** | **29.46±0.47** | **58.03±0.34** |

To further comprehensively verify the effectiveness of our proposed consistency learning strategy, we conduct additional experiments on two more datasets beyond Table 2. As clearly shown in Table 10, the reported results consistently show that our consistency learning significantly outperforms both MSE and InfoNCE across all evaluated datasets. This superior overall performance strongly highlights the ability of our method to effectively capture high-order semantic similarity relations via the powerful distribution-based alignment. The reason why the two compared metrics perform poorly may be that MSE only focuses on point-wise errors and ignores inter-sample correlations, while InfoNCE is prone to instability due to the influence of negative sample selection. Overall, these additional results provide a broader and more empirical basis for our conclusions, further reinforcing the superiority, and robustness of our proposed consistency learning approach.

## I  ADDITIONAL SENSITIVITY ANALYSIS

Here we extend the sensitivity analysis of the key hyper-parameters $K$, $M$ and $\alpha$ to three additional datasets. The results in Figure 6 show that the clustering performance on all datasets increases slowly as $K$ ranges from 32 to 128, but starts to decline when $K$ becomes too large. This is consistent with the observations on the two datasets in Figure 4 of the main text, indicating that a too high-dimensional subspace introduces redundant or irrelevant information, which hinders the compactness of node embeddings. For the number of anchor samples $M$, the model's performance is poor when $M$ is very small, due to the insufficient anchor samples to represent the neighborhood structure of nodes. As $M$ increases, the performance improves and remains stable, which aligns with the findings in the main text. For the teleport probability $\alpha$, the model performance is stable at low $\alpha$ values but deteriorates with further increases, showing a marked decline at $\alpha$=0.8. This trend aligns with findings in the main text. The degradation occurs because a higher $\alpha$ results in a more localized diffusion process, which over-prioritizes the node's own information. Conversely, a smaller $\alpha$ promotes global integration of graph information, which is essential for learning diverse and complementary representations. These consistent trends across different datasets further validate the robustness of our model with respect to the three hyper-parameters.

## J  GRAPH CONVOLUTIONAL FILTER $v.s.$ GCN

To show that our proposed disentangling of graph convolutional filters (GCF) and weight matrices can enhance the robustness of the model, we conduct comparative experiments between GCF+MLP

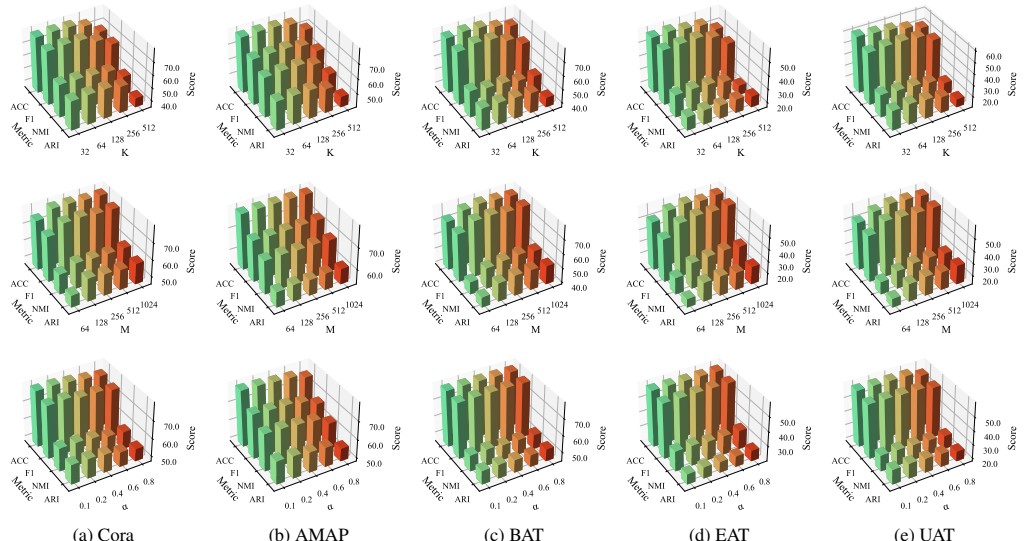

Figure 6: The sensitivity analysis of three hyper-parameters on all five datasets.

and GCN under the CoCo framework as the number of convolutional layers (i.e., the parameter $t$ in GCF, Equation 1) increased. The results are shown in Figure 7. It can be observed that GCF+MLP consistently outperforms GCN in terms of clustering performance across different convolutional layers. We also note that the performance of GCF+MLP remains relatively stable as the number of lay-

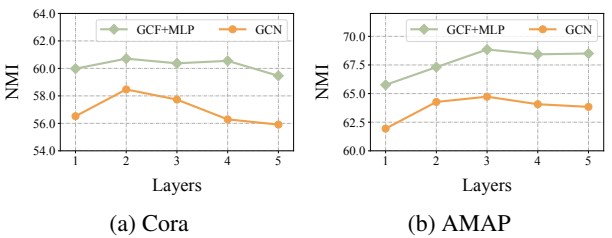

Figure 7: Robustness analysis.

ers ($t$ in Equation 1) increases, whereas GCN tends to first improve and then deteriorate with the addition of more layers in most cases. This evidences the high performance and robustness brought about by disentangling the GCF from the weight matrices within our CoCo framework.

## K  RELATED WORK

### K.1  DEEP GRAPH CLUSTERING

Over the years, substantial research efforts have been devoted to advancing deep graph clustering, with the goal of partitioning nodes in a graph into cohesive and well-connected clusters in an end-to-end framework (Liu et al., 2022b). Deep learning breakthroughs have enabled the development of graph algorithms that leverage both graph structures and node attributes to tackle this inherently complex task (Shi et al., 2019; Cheng et al., 2021; Yang et al., 2023; Yi et al., 2023; Liu et al., 2024b; Trivedi et al., 2024; Guo et al., 2025; Ren et al., 2025; Wang et al., 2025b). Specifically, SDCN (Bo et al., 2020) and DFCN (Tu et al., 2021) provide a foundation for integrating graph structure into clustering objectives, laying the groundwork for more recent models. Several advanced methods have since been proposed, each targeting specific challenges in graph clustering. AGCN (Peng et al., 2021) designs a dynamic attention mechanism to adaptively aggregate the multi-scale features, while MCGC (Pan & Kang, 2021) learns a consensus graph by filtering out high-frequency noise, preserving graph geometric features. GLCC (Ju et al., 2023) is the first to extend clustering tasks to the graph-level scenario using the concept of contrastive learning. To address the challenges of large-scale graph data, Dink-Net (Liu et al., 2023b) integrates representation learning with clustering optimization through dilation-shrink mechanisms and uses mini-batches for scalability. To eliminate the need for a predefined cluster number, RGC (Liu et al., 2023a) employs reinforcement learning to jointly learn representations and adaptively adjust cluster numbers for bet-

ter cohesion and separation. To generalize graph clustering to heterophilic scenarios, DGCN (Pan & Kang, 2023) and DMGC (Guo et al., 2025) reconstructs homophilic and heterophilic graphs with a mixed filter that captures multi-frequency features. Additionally, GCLR (Trivedi et al., 2024) and MARK (Fu et al., 2025) leverage large language models to enhance text-attributed graph clustering. Despite these advancements, a common challenge remains: the inability to capture long-range dependencies in graphs and the neglect of inherent redundant information and noise in data. These factors are crucial to capturing cohesive clusters in complex graph structures. By learning robust and compact node representations in our method CoCo, it is promising to uncover meaningful patterns and communities within the graph.

## K.2 GLOBAL DEPENDENCIES IN GNNS

Despite the success of GNNs in various graph-structured applications, their message-passing mechanism (Gilmer et al., 2017) usually restricts them from capturing long-range dependencies. To overcome this, a growing body of research has been dedicated to enhancing GNNs with global dependency capabilities, which can be roughly categorized into two lines: *implicit global interaction* and *explicit structural augmentation*. The first group keeps the original graph unchanged and introduces model-level global interactions, enabling node representations to exchange information globally in feature space (Ying et al., 2021; Zhang et al., 2022; Cai et al., 2023; Liang et al., 2023; Xing et al., 2024; Yi et al., 2025a). For example, Graphormer (Ying et al., 2021) leverages structural encodings within a Transformer framework to effectively capture graph dependencies, while CoBFormer (Xing et al., 2024) adopts a bi-level architecture and collaborative training to more effectively capture global dependencies while retaining important local information. The second group focuses on directly modifying the graph's connectivity by adding 'shortcuts' to physically shorten the distance between distant nodes, enabling standard GNNs to more flexibly capture global relationships (Wu et al., 2022; Guo et al., 2024; Ju et al., 2024a; Yi et al., 2025b; Li et al., 2025). Specifically, GraphEdit (Guo et al., 2024) leverages large language models to learn global node-wise dependencies, denoise connections, and enhance graph structure learning through instruction-tuned reasoning. In our work, we follow the second paradigm by constructing a graph diffusion matrix to capture global dependencies, effectively modeling indirect connections and enabling clusters to accurately represent underlying community structures.

## K.3 NOISE-RESILIENT LEARNING IN GNNS

GNNs are highly sensitive to various types of noise (Ju et al., 2024b; 2025), including feature perturbations and redundant information, which can distort feature distributions and degrade performance. Existing noise-resilient learning approaches can be broadly divided into two lines: *adversarial defense* and *loss refinement*. The first line explicitly defends against malicious or worst-case perturbations by simulating or constraining adversarial noise (Feng et al., 2019; Alchihabi et al., 2023; Yuan et al., 2023a). For instance, BVAT (Deng et al., 2023) employs batch virtual adversarial training to counteract noise and better capture local graph structures, while GCORN (Abbahaddou et al., 2024) enhances robustness against node feature attacks by enforcing orthonormal weight matrices. The second line enhances robustness by modifying training objectives to handle noisy or redundant inputs (Wang & Yang, 2022; Huo et al., 2023; Yuan et al., 2023b; Ju et al., 2026). Specifically, BRGCL (Wang & Yang, 2022) iteratively estimates confident nodes to compute robust cluster prototypes and applies prototypical contrastive learning to learn noise-resistant node representations, while T2-GNN (Huo et al., 2023) uses a dual teacher-student framework to transfer clean patterns and recover noisy or corrupted node attributes. Unlike adversarial or loss refinement approaches, we introduce the idea of low-rank learning to help the model learn compact representations for graph clustering, as it explicitly models the underlying low-dimensional structure, and separate noise and redundancy at the data-distribution level, thereby producing more robust and more clustering-friendly representations.

