# OpenReview forum: "Compactness and Consistency: A Conjoint Framework for Deep Graph Clustering"
_ICLR.cc/2026/Conference — ICLR 2026 Oral_

### Official Review · Reviewer_3Q5j · 2025-10-27

**Soundness:** 2
**Presentation:** 3
**Contribution:** 2
**Rating:** 4
**Confidence:** 3

**Summary:**

This paper introduces CoCo, a conjoint framework for deep graph clustering. Addressing limitations of Graph Neural Networks (GNNs) in capturing global relationships and handling redundancy and noise in graph data, CoCo aims to learn node representations endowed with compactness and consistency. The method first leverages graph convolutional filters to extract node features from both local and global views. These representations are then encoded into low-rank compact embeddings using a Gaussian Mixture Model (GMM) to eliminate redundancy and noise. Finally, a consistency learning strategy is developed to facilitate knowledge transfer between the compact embeddings from the two perspectives, thereby enriching node semantics. Experimental results demonstrate that CoCo outperforms state-of-the-art methods on various benchmark datasets.

**Strengths:**

The paper clearly identifies existing GNN limitations in capturing global relationships and handling redundancy/noise, proposing targeted solutions. The proposed CoCo framework innovatively combines graph diffusion (for global view), disentangled graph convolutional filters (for feature extraction), GMM-based low-rank compactness learning, and cross-view consistency learning, with these components synergistically improving clustering performance.

**Weaknesses:**

The paper states that GMM aims to maximize p(Z|\lambda) and that prior probabilities and covariance matrices are fixed. A more detailed explanation is needed for this simplification, such as why these parameters are fixed and how this impacts the model's generality. The limitations of the existing methods lack support from newer literature. Related work does not correspond to the limitations of the existing works. Theorem 2 requires further explanation. The information loss or over-smoothing risk that low-pass filtering and low-rank approximation may bring during denoising may reduce the practical application of this paper.

**Questions:**

Theorem 2 seems to be insufficiently proven.

Is the setting of noise only beneficial to the author's method?

---

> ### Author Response · Authors · 2025-11-19
>
> We are truly grateful for the time you have taken to review our paper and your insightful review. Here we address your comments in the following, where Q denotes the comment of the reviewer and R denotes our response.
>
> > Q1. The paper states that GMM aims to maximize p(Z|\lambda) and that prior probabilities and covariance matrices are fixed. A more detailed explanation is needed for this simplification, such as why these parameters are fixed and how this impacts the model's generality.
>
> R1. Thanks for your valuable suggestion! In our low-rank subspace training, the quality of the means $\boldsymbol \lambda_{\cdot,k}$ and the posteriors $\gamma(y_{jk})$ are the most crucial factors. Therefore, we fix the prior probabilities (or mixture weights) to be equal and the covariance matrices to be isotropic. It does not impact the model's generality and offers certain advantages.: **(1)** on the one hand, the prior does not affect the overall trend of the posterior distribution, which is mainly determined by data fitting; on the other hand, fixing equal priors helps mitigate cluster collapse and encourages the learned subspace to cover the embedding space more uniformly. **(2)** Moreover, fixing the covariance matrices forces the model's fitting capacity to concentrate on the ''mean-defined subspace''. This is because, under uniform mixture weights and isotropic covariances, and based on the negative ELBO bound in the M-step, maximizing the log-likelihood is equivalent to minimizing the weighted squared–distance objective $\sum_{j,k} \gamma(y_{jk})\|\|\mathbf z_{\cdot,j} - \boldsymbol \lambda_{\cdot,k}\|\|^2.$
>
> **Please refer to the blue-highlighted content above Remark 1 in Section 2.2, in the revised manuscript.**
>
> > Q2. The limitations of the existing methods lack support from newer literature. Related work does not correspond to the limitations of the existing works.
>
> R2. Thanks for your valuable suggestion! We have supplemented the Introduction (Section 1) with the latest literature and highlighted the shortcomings of existing works for each limitation. Additionally, I have rewritten the Related Work section (Appendix K) so that each topic directly corresponds to the limitations of existing studies.
>
> **Please refer to the blue-highlighted content in Section 1, Appendix K in the revised manuscript.**
>
> > Q3. Theorem 2 requires further explanation. Theorem 2 seems to be insufficiently proven.
>
> R3. Thanks for your valuable suggestion! Before and after Theorem 2, we have added explanations regarding low-rank preservation and the intuitive meaning and significance of Theorem 2's results. Specifically, Theorem 2(1) guarantees the invariance of each individual's total signal mass. This is crucial for fairness and interpretability, as it prevents the model from systematically biasing the reconstructed profiles of any individual; while Theorem 2(2) ensures that our reconstruction prioritizes the retention of the significant variations (with large $z_{ij}$) due to the cross-term $\sum_{i,j}z_{ij} \hat{z}_{ij}$. This enables our low-rank reconstruction to align strongly with these dominant patterns. Additionally, the original proof of Theorem 2 is valid and justified. To enhance the completeness and readability of the proof, we have provided a more detailed and rigorous elaboration in Appendix B.3.
>
> **Please refer to the blue-highlighted content in Feature Reconstruction of Section 2.2 and Appendix B.3 in the revised manuscript**.

---

> > ### Author Response · Authors · 2025-11-19
> >
> > > Q4. The information loss or over-smoothing risk that low-pass filtering and low-rank approximation may bring during denoising may reduce the practical application of this paper.
> >
> > R4. Thanks for your valuable question! **Regarding low-pass filtering**, its purposes are: (1) to suppress high-frequency noise components, and (2) to decouple the graph convolution operation from the weight matrix while integrating both attribute and structural information (Eqs. (1) and (2)). Although some information loss is inevitable, it primarily filters out noisy components. Experimentally, the superior clustering performance of our CoCo method, as shown in Tables 1 and 9, demonstrates that our approach can effectively reveal the true distribution of graph data. Furthermore, as illustrated in Appendix J by comparing low-pass filter + MLP with GCN, our proposed decoupling strategy exhibits better resistance to over-smoothing, i.e., as the number of layers increases, low-pass filter + MLP experiences performance degradation later or maintains a more stable trend compared to GCN.
> >
> > **Regarding low-rank learning**, besides ensuring low-rank property (as stated above Theorem 2), our derived Theorem 2 further shows that our low-rank reconstruction guarantees the invariance of each individual's total signal mass. This property is crucial for fairness and interpretability, as it prevents the model from systematically biasing the reconstructed profiles of any individual. At the same time, it ensures that our reconstruction prioritizes the retention of significant variations (i.e., large $z_{ij}$) through the cross-term $\sum_{i,j} z_{ij} {z}'_{ij}$, enabling the low-rank reconstruction to align strongly with these dominant patterns. Moreover, at the end of Section 2.2, we also tune the low-rank representations, which combine the global trends captured by the low-rank component with the local details preserved in the original embeddings, thereby mitigating the over-smoothing that may result from relying solely on a low-rank constraint and preventing model collapse.
> >
> > **Please refer to the blue-highlighted content in Tables 1 and 9, Appendix J, and the end of Section 2.2 in the revised manuscript**.
> >
> > > Q5. Is the setting of noise only beneficial to the author's method?
> >
> > R5. Thanks for your valuable suggestion! **First**, in our main experimental results (Tables 1 and 8), we do not add any additional noise to the publicly available benchmark datasets, which already contain intrinsic noise. From the numerical comparisons under various metrics, it is evident that our method outperforms the 19 competitive baselines. **Second**, in Section 3.6, we further evaluate performance on the Cora dataset by introducing additional noise. The noise considered here is common and uncorrelated with our method, i.e., Gaussian feature noise and Bernoulli structural noise. As shown in Table 3, compared with competitive methods such as GraphLearner and MAGI, our approach demonstrates stronger robustness, achieving superior clustering performance under these noisy conditions.
> >
> > In light of these responses, we hope we have addressed your concerns, and hope you could consider raising your score. If there are any additional notable points of concern that we have not yet addressed, please do not hesitate to share them, and we will promptly attend to those points.

---

> > > ### Comment · Reviewer_3Q5j · 2025-11-25
> > >
> > > Thank you for your reply. I will increase my score.

---

> > > > ### Author Response · Authors · 2025-11-25
> > > >
> > > > Dear Reviewer 3Q5j,
> > > >
> > > > Thanks for your feedback and increasing the rating! We are pleased to know that our responses have addressed your concerns. We will add the rebuttal contents to the main paper in the final version following your valuable suggestions.
> > > >
> > > > Best regards,
> > > >
> > > > the Authors

---

### Official Review · Reviewer_4B3S · 2025-10-29

**Soundness:** 4
**Presentation:** 4
**Contribution:** 4
**Rating:** 8
**Confidence:** 4

**Summary:**

This paper proposes a novel framework for deep graph clustering that aims to learn node representations which are both compact and consistent. The model addresses two key limitations in existing methods: the inability of standard GNNs to capture global node relationships and the sensitivity to noise and redundancy in graph data. The methodology first extracts node features from both local (original adjacency matrix) and global (graph diffusion matrix) views using disentangled graph convolutional filters. These features are then encoded into a low-rank subspace via a Gaussian mixture model to eliminate redundancy and noise, producing compact embeddings. Finally, a consistency learning strategy aligns the similarity distributions of nodes between the two views using a symmetric KL-divergence loss, enriching the semantics without needing negative samples. Extensive experiments on benchmark datasets show that CoCo outperforms state-of-the-art methods.

**Strengths:**

- The model effectively integrates both local and global structural information to form comprehensive node representations
- The low-rank compactness learning effectively reduces noise and redundancy, leading to more robust embeddings.
- The authors provide a detailed theoretical analysis that justifies the rationale behind the proposed technique.
- It demonstrates strong generalization capability by achieving superior performance across diverse tasks, including both node clustering and classification.
- The approach is computationally efficient and scalable to large graphs, owing to its linear complexity.

**Weaknesses:**

- The GMM-based low-rank learning adds model complexity and training steps.
- The sensitivity analysis of hyperparameters is relatively limited.

**Questions:**

- In Equation 1, what is the impact of different values of $k$ on model performance? Please provide further experimental explanation.
- The author's original intention to propose the use of graph convolutional filters is to disentangle the filters and the weight matrix. How to prove its effectiveness?
- In Section 2.2, why is GMM used for low-rank subspace training? Why is the original embedding representation $\mathbf{Z}$ added in Equation 5?
- Since the EM algorithm is independent of the model training, how does its initialization affect the model? What is the additional training cost required?

---

> ### Author Response · Authors · 2025-11-19
>
> We are truly grateful for the time you have taken to review our paper and your insightful review. Here we address your comments in the following, where Q denotes the comment of the reviewer and R denotes our response.
>
> > Q1. The author's original intention to propose the use of graph convolutional filters is to disentangle the filters and the weight matrix. How to prove its effectiveness?
>
> R1. Thanks for your valuable problem! Indeed, we have provided a detailed analysis of this in Appendix K. It can be observed that GCF+MLP consistently outperforms GCN in terms of clustering performance across different convolutional layers. We also note that the performance of GCF+MLP remains relatively stable as the number of layers ($t$ in Eq. 1) increases, whereas GCN tends to first improve and then deteriorate with the addition of more layers in most cases. This evidences the high performance and robustness brought about by disentangling the GCF from the weight matrices within our framework.
>
> **Please refer to the blue-highlighted content in Appendix J, along with Figure 7 in the revised manuscript.**
>
> > Q2. In Section 2.2, why is GMM used for low-rank subspace training? Why is the original embedding representation Z added in Equation 5?
>
> R2. Thanks for your valuable question! To implement low-rank learning, we utilize GMM trained with the EM algorithm for subspace modeling. Its primary role is to reduce dimensionality while preserving key information. Its advantage lies in having linear modeling complexity. In the experiment, we verify that the subspace searching algorithm can achieve good performance within 10 iterations across different datasets, incurring negligible additional computational cost (Section 3.8). We also compare it with other dimensionality reduction algorithms in Appendix C, including PCA, k-means, and autoencoder. Then we reconstruct the features based on the posterior probabilities and the learned subspace. We theoretically prove in Theorem 2 that the proposed approach reconstructs the embedding $\hat{\mathbf Z}$ in a way that optimally preserves individual information and total variation of the original embedding $\mathbf Z$.
>
> Compared with gradient-based optimization, the EM algorithm provides a more stable procedure with stronger convergence guarantees for low-rank estimation. However, EM-based low-rank learning operates outside the gradient flow. Thus, the first role of Eq. (5) is to inject the low-rank representation back into the gradient path, ensuring trainability and also helping avoid model collapse. Moreover, the low-rank component captures global trends while the original $\mathbf Z$ preserves local details, so Eq. (5) also mitigates the over-smoothing that a purely low-rank constraint might induce (see the end of Section 2.2).
>
> **Please refer to the blue-highlighted content in Section 3.8, Appendix C, and the end of Section 2.2 in the revised manuscript.**
>
> > Q3. Since the EM algorithm is independent of the model training, how does its initialization affect the model? What is the additional training cost required?
>
> R3. Thanks for your valuable problem! In our framework, the number of EM iterations is set to 10 by default. Here we examine how different iteration counts affect performance (using NMI as an example) and efficiency (EM time per iteration / total time per iteration), with the results presented in the table below.
>
> | Iterations | Cora |           | AMAP |           |
> |------------|-----------|-----------|-----------|-----------|
> |              | NMI      | Efficiency   | NMI       | Efficiency   |
> | 1         | 47.99±4.07     | 1.62%        | 62.96±2.16     | 2.23%        |
> | 5         | 59.81±0.73     | 2.93%        | 68.50±1.25     | 3.27%        |
> | 10         | 60.71±0.59     | 4.28%        | 68.85±1.55     | 4.24%        |
> | 20         | 60.69±0.57     | 6.69%        | 68.87±1.88     | 6.16%        |
> | 30         | 60.71±0.60     | 8.46%        | 68.81±1.70     | 8.42%        |
>
> From the results, we can see that although the proportion of EM iteration time in the total runtime gradually increases with the number of iterations, the performance stabilizes after 10 iterations, indicating that the model has already converged. Further increasing the number of EM iterations yields no additional benefits. Therefore, it can be concluded that the EM iterative algorithm accounts for only about 4% of the total training time, demonstrating remarkably high efficiency.
>
> **Please refer to the blue-highlighted content in Section 3.8, along with the additions to Table 6 in the revised manuscript.**
>
> Thanks again for appreciating our work and for your constructive suggestions. Please let us know if you have further questions.

---

> > ### Comment · Reviewer_4B3S · 2025-11-26
> >
> > Thanks for your responces. My concerns are well addressed. I will maintain my positive rating.

---

> > > ### Author Response · Authors · 2025-11-26
> > >
> > > Dear Reviewer 4B3S,
> > >
> > > We are pleased to know that our responses have addressed your concerns. Please do not hesitate to reach out to us if you have any further questions
> > >
> > > Best regards,
> > >
> > > the Authors

---

### Official Review · Reviewer_y7YE · 2025-10-29

**Soundness:** 4
**Presentation:** 4
**Contribution:** 4
**Rating:** 8
**Confidence:** 5

**Summary:**

This paper addresses the limitations of graph clustering, particularly their inability to capture global node relations and their susceptibility to redundancy and noise in learned representations. The authors propose a conjoint framework, CoCo, which learns robust representations from local and global views through graph convolutional filters, encodes them into low-rank compact embeddings to eliminate redundancy and noise, and further introduces a consistency learning strategy to transfer knowledge across views. Extensive experiments on benchmark datasets demonstrate that CoCo consistently outperforms state-of-the-art methods.

**Strengths:**

- The research topic is important and meaningful in the graph learning field, and appeals to a broad audience.
- The paper is well-written and easy to follow. Its cutting-edge, rational technical solution addresses long-distance dependence, redundancy, and noise in feature representation, which are often overlooked in the current graph clustering field.
- The theoretical analysis further reinforces the soundness and validity of the proposed scheme.
- Extensive experiments and analyses across multiple downstream tasks and over ten baseline methods fully demonstrate the proposed method’s superiority, and effectively validate the study’s stated motivations.

**Weaknesses:**

- Neither the Introduction section nor the methods selected for comparative experiments have incorporated studies published in 2025.
- The main research results lack statistical significance tests.

**Questions:**

- For the graph convolutional filter adopted in Eq. 1, how well would it perform if this filter were replaced with GCN?
- The authors proposed GMM-based compactness learning; what impact would using only PCA dimensionality reduction have?
- In Eq. 7, the authors only use sum averaging for the semantic representation of the two different branches. Would performance improve if attention weighting were used instead?
- For the node classification evaluation, clarification is needed on whether the authors applied a classification head to the representation and trained via a supervised loss, or trained with an unsupervised loss before supervised fine-tuning.
- The calculations of $\mathbf{p}$ and $\mathbf{q}$ are based on cosine similarity. Why was this similarity metric chosen instead of other methods? How would the results compare with those from other methods?

---

> ### Author Response · Authors · 2025-11-19
>
> We are truly grateful for the time you have taken to review our paper and your insightful review. Here we address your comments in the following, where Q denotes the comment of the reviewer and R denotes our response.
>
> > Q1. For the graph convolutional filter adopted in Eq. 1, how well would it perform if this filter were replaced with GCN?
>
> R1. Thanks for your valuable suggestion! Indeed, we have provided a detailed analysis of this in Appendix K. It can be observed that GCF+MLP consistently outperforms GCN in terms of clustering performance across different convolutional layers. We also note that the performance of GCF+MLP remains relatively stable as the number of layers ($t$ in Eq. 1) increases, whereas GCN tends to first improve and then deteriorate with the addition of more layers in most cases. This evidences the high performance and robustness brought about by disentangling the GCF from the weight matrices within our framework.
>
> **Please refer to the blue-highlighted content in Appendix J, along with Figure 7 in the revised manuscript.**
>
> > Q2. The authors proposed GMM-based compactness learning; what impact would using only PCA dimensionality reduction have?
>
> R2. Thanks for your valuable problem! As discussed in the first part of Appendix C, the advantage of GMM lies in its linear modeling complexity, with time and space complexities of $O(NDK)$ and $O(ND)$, respectively, while PCA has time and space complexities of $O(ND^2+D^3)$ and $O(ND+D^2)$. When the number of features is high, the complexity of PCA is also high.
>
> **Please refer to the blue-highlighted content in Appendix C in the revised manuscript.**
>
> > Q3. In Eq. 7, the authors only use sum averaging for the semantic representation of the two different branches. Would performance improve if attention weighting were used instead?
>
> R3. Thanks for your valuable suggestion! We compare the two fusion methods across all datasets, with results shown in the table below. It can be observed that both ways have their respective advantages on different datasets, with only marginal differences in performance, making them nearly equivalent in overall effectiveness. However, simple sum averaging is more efficient in terms of both time and space complexity, which is why we have adopted this approach in our framework.
>
> | Dataset | Metric | Attention Fusion    | Sum Averaging (Ours) |
> |---------|--------|-------------|-------------|
> | Cora    | ACC    | 79.10±0.62  | **79.36±0.69** |
> |         | NMI    | 60.58±0.53  | **60.71±0.59** |
> |         | ARI    | 58.71±1.46  | **58.76±1.47** |
> |         | F1     | 77.65±0.50  | **77.95±0.72** |
> | AMAP    | ACC    | **79.31±0.98**  | 79.27±0.70 |
> |         | NMI    | **68.85±1.85**  | **68.85±1.55**    |
> |         | ARI    | **61.07±2.34**  | 60.94±1.51 |
> |         | F1     | **72.63±1.31** | 72.36±1.15    |
> | BAT     | ACC    |**78.93±0.78**  | 78.85±0.91 |
> |         | NMI    | 54.68±0.75  | **55.00±0.87** |
> |         | ARI    | 53.32±1.00  | **53.52±1.15** |
> |         | F1     | **78.60±0.92**  | 78.56±1.01 |
> | EAT     | ACC    | 58.65±0.64 | **58.87±0.49** |
> |         | NMI    | 33.70±1.43  |  **34.10±1.26**    |
> |         | ARI    | 27.76±2.11  |  **27.91±1.52**    |
> |         | F1     | 57.58±1.97  | **58.06±2.64** |
> | UAT     | ACC    | **59.86±0.45**  | 59.68±0.36 |
> |         | NMI    | 29.55±0.58  | **30.12±0.51** |
> |         | ARI    | 29.02±1.32  | **29.46±0.47** |
> |         | F1     | **58.28±0.38**  | 58.03±0.34 |
>
> > Q4. For the node classification evaluation, clarification is needed on whether the authors applied a classification head to the representation and trained via a supervised loss, or trained with an unsupervised loss before supervised fine-tuning.
>
> R4. Sorry for the confusion. We first perform unsupervised pre-training using the proposed framework, followed by supervised fine-tuning on a labeled dataset. We have clarified this procedure more explicitly in the revised manuscript.
>
> **Please refer to the blue-highlighted content in Section 3.9 in the revised manuscript.**

---

> > ### Author Response · Authors · 2025-11-19
> >
> > > Q5. The calculations of p and q are based on cosine similarity. Why was this similarity metric chosen instead of other methods? How would the results compare with those from other methods?
> >
> > R5. Thanks for your valuable suggestion! Here, we additionally select the Gaussian kernel and Pearson correlation coefficient as similarity measures to compute $\mathbf{p}$ and $\mathbf{q}$. The results on the Cora and AMAP datasets are shown in the table below.
> >
> > | Dataset | Metric | Gaussian   | Pearson | Cosine (Ours) |
> > |---------|--------|-------------|-------------|-------------|
> > | Cora  | ACC   | 76.41±1.56  | 76.92±0.75 | **79.36±0.69** |
> > |           | NMI    | 59.49±1.93  | 58.59±0.96 | **60.71±0.59** |
> > |           | ARI    | 56.29±2.78  | 55.58±2.02 | **58.76±1.47** |
> > |           | F1     | 73.53±2.78  | 75.38±1.33 | **77.95±0.72** |
> > | AMAP    | ACC    | 78.99±0.53  | 79.06±0.84 | **79.27±0.70** |
> > |           | NMI    | 68.69±1.26  | 68.57±1.09    | **68.85±1.55**    |
> > |           | ARI    | 60.37±1.55 | 60.77±1.36 | **60.94±1.51** |
> > |           | F1     | 72.12±0.69 | 72.18±0.96    | **72.36±1.15**    |
> >
> > As can be seen from the results, cosine similarity is optimal aross two datasets, while the other two metrics perform highly unstable across different datasets, and this holds true for other datasets as well. Overall, cosine similarity is computationally efficient and achieves the best performance.
> >
> > Thanks again for appreciating our work and for your constructive suggestions. Please let us know if you have further questions.

---

> > > ### Comment · Reviewer_y7YE · 2025-11-26
> > >
> > > Thank you for your efforts in resolving my concerns. All my concerns have been resolved, and I maintain my score.

---

> > > > ### Author Response · Authors · 2025-11-26
> > > >
> > > > Dear Reviewer y7YE,
> > > >
> > > > We are pleased to know that our responses have addressed your concerns. Please do not hesitate to reach out to us if you have any further questions
> > > >
> > > > Best regards,
> > > >
> > > > the Authors

---

### Official Review · Reviewer_wkWL · 2025-10-30

**Soundness:** 4
**Presentation:** 4
**Contribution:** 4
**Rating:** 8
**Confidence:** 5

**Summary:**

This paper proposes a conjoint framework named CoCo for deep graph clustering, which captures compactness and consistency in learned node representations by leveraging graph convolutional filters for local-global view feature extraction, GMM-based low-rank embedding for redundancy elimination, and consistency learning for semantic enhancement. Experimental results on multiple benchmark datasets show CoCo outperforms state-of-the-art methods in graph clustering and demonstrates good scalability and robustness.

**Strengths:**

1.The paper is clearly written and logically rigorous, presenting the proposed method in a clear way.

2.The novelty and technical solution of the paper is sound and reasonable, long-range dependence and redundancy/noise are indeed the points that are easily ignored in graph clustering.

3.The paper provides rigorous theoretical analysis and well-defined theoretical framework, Theorems 1-3 provide the optimal filtering parameters and the convergence guarantee for the dimension reduction, respectively.

4.The authors conduct extensive experiments for the proposed framework, showing the superiority of the overall results. The ablation study also verifies the effectiveness of each component, and the analysis on different tasks and different scenarios also illustrates the good generalization of the framework.

**Weaknesses:**

1.The details regarding compactness learning and the EM algorithm are not elaborated or analyzed with sufficient clarity.

2.The motivation behind compact learning for redundancy reduction is not sufficiently clear.

3.The fusion of representations from different branches may lack flexibility.

**Questions:**

1.Although this paper claims a linear time complexity, could the iteration count of the EM algorithm impact the overall training efficiency?

2.Why is EM-based subspace learning used?

3.Why compact learning can eliminate redundancy and mitigate the impact of noise requires further clarification.

4.When fusing local and global representations, equal weights are used. Why not adopt adaptive weights (e.g., learned via attention) to prioritize more informative views?

---

> ### Author Response · Authors · 2025-11-19
>
> We are truly grateful for the time you have taken to review our paper and your insightful review. Here we address your comments in the following, where Q denotes the comment of the reviewer and R denotes our response.
>
> > Q1. Although this paper claims a linear time complexity, could the iteration count of the EM algorithm impact the overall training efficiency?
>
> R1. Thanks for your valuable suggestion! In our framework, the number of EM iterations is set to 10 by default. Here we examine how different iteration counts affect performance (using NMI as an example) and efficiency (EM time per iteration / total time per iteration), with the results presented in the table below.
>
> | Iterations | Cora |           | AMAP |           |
> |------------|-----------|-----------|-----------|-----------|
> |              | NMI      | Efficiency   | NMI       | Efficiency   |
> | 1         | 47.99±4.07     | 1.62%        | 62.96±2.16     | 2.23%        |
> | 5         | 59.81±0.73     | 2.93%        | 68.50±1.25     | 3.27%        |
> | 10         | 60.71±0.59     | 4.28%        | 68.85±1.55     | 4.24%        |
> | 20         | 60.69±0.57     | 6.69%        | 68.87±1.88     | 6.16%        |
> | 30         | 60.71±0.60     | 8.46%        | 68.81±1.70     | 8.42%        |
>
> From the results, we can see that although the proportion of EM iteration time in the total runtime gradually increases with the number of iterations, the performance stabilizes after 10 iterations, indicating that the model has already converged. Further increasing the number of EM iterations yields no additional benefits. Therefore, it can be concluded that the EM iterative algorithm accounts for only about 4% of the total training time, demonstrating remarkably high efficiency.
>
> **Please refer to the blue-highlighted content in Section 3.8, along with the additions to Table 6 in the revised manuscript.**
>
> > Q2. Why is EM-based subspace learning used?
>
> R2. Thanks for your valuable questions! **First**, low-rank representations can effectively exploit the inherent underlying correlation structure among data and suppress the impact of noise under the assumption that high-dimensional data points often intrinsically lie on a low-dimensional subspace. In graph clustering tasks, we aim to utilize low-rank subspace and representation learning to explore representations that excellently reflect the inherent distribution of graph data, thereby achieving better clustering performance when inputting the representations into a clustering algorithm. **Second**, to implement low-rank learning, we utilize a Gaussian mixture model trained with the EM algorithm for subspace modeling. Its primary role is to reduce dimensionality while preserving key information, and its advantage lies in having linear modeling complexity. We have also compared it with other dimensionality reduction algorithms, including PCA, k-means, and autoencoder.
>
> **Please refer to the blue-highlighted content in the first paragraph of Section 2.2 and Appendix C in the revised manuscript.**
>
> > Q3. Why compact learning can eliminate redundancy and mitigate the impact of noise requires further clarification.
>
> R3. Thanks for your valuable questions! In our framework, compact learning is achieved through a low-rank modeling procedure built upon the learned embeddings. Specifically, we fit a GMM on the embedding space and train it via the EM algorithm, where the mean vectors of the learned Gaussian components serve as the basis vectors of a low-dimensional subspace. The data are then reconstructed from these subspace vectors, yielding a matrix whose rank is inherently bounded by the dimension of the learned subspace. This compact representation effectively eliminates redundancy, because only the principal, cluster-level directions captured by the GMM means are retained while correlated or weakly informative variations are removed during the projection & reconstruction process. Moreover, the low-rank reconstruction naturally suppresses noise, as high-frequency or unstable fluctuations in the original embeddings cannot be expressed within the constrained subspace and thus vanish when reconstructing onto the original dimensions. Consequently, the resulting compact representation preserves the essential structure while filtering out redundant and noisy components.

---

> > ### Author Response · Authors · 2025-11-19
> >
> > > Q4. When fusing local and global representations, equal weights are used. Why not adopt adaptive weights (e.g., learned via attention) to prioritize more informative views?
> >
> > R4. Thanks for your valuable suggestion! We compare the two fusion methods (Attention Fusion v.s. Sum Averaging) across all datasets, with results shown in the table below. It can be observed that both ways have their respective advantages on different datasets, with only marginal differences in performance, making them nearly equivalent in overall effectiveness. However, simple sum averaging is more efficient in terms of both time and space complexity, which is why we have adopted this approach in our framework.
> >
> > | Dataset | Metric | Attention Fusion    | Sum Averaging (Ours) |
> > |---------|--------|-------------|-------------|
> > | Cora    | ACC    | 79.10±0.62  | **79.36±0.69** |
> > |         | NMI    | 60.58±0.53  | **60.71±0.59** |
> > |         | ARI    | 58.71±1.46  | **58.76±1.47** |
> > |         | F1     | 77.65±0.50  | **77.95±0.72** |
> > | AMAP    | ACC    | **79.31±0.98**  | 79.27±0.70 |
> > |         | NMI    | **68.85±1.85**  | **68.85±1.55**    |
> > |         | ARI    | **61.07±2.34**  | 60.94±1.51 |
> > |         | F1     | **72.63±1.31** | 72.36±1.15    |
> > | BAT     | ACC    |**78.93±0.78**  | 78.85±0.91 |
> > |         | NMI    | 54.68±0.75  | **55.00±0.87** |
> > |         | ARI    | 53.32±1.00  | **53.52±1.15** |
> > |         | F1     | **78.60±0.92**  | 78.56±1.01 |
> > | EAT     | ACC    | 58.65±0.64 | **58.87±0.49** |
> > |         | NMI    | 33.70±1.43  |  **34.10±1.26**    |
> > |         | ARI    | 27.76±2.11  |  **27.91±1.52**    |
> > |         | F1     | 57.58±1.97  | **58.06±2.64** |
> > | UAT     | ACC    | **59.86±0.45**  | 59.68±0.36 |
> > |         | NMI    | 29.55±0.58  | **30.12±0.51** |
> > |         | ARI    | 29.02±1.32  | **29.46±0.47** |
> > |         | F1     | **58.28±0.38**  | 58.03±0.34 |
> >
> > Thanks again for appreciating our work and for your constructive suggestions. Please let us know if you have further questions.

---

### Official Review · Reviewer_bn3i · 2025-10-31

**Soundness:** 4
**Presentation:** 4
**Contribution:** 3
**Rating:** 6
**Confidence:** 5

**Summary:**

This paper introduces a compactness-and-consistency framework designed for deep graph clustering. The contributions are three-fold as follows:
1.The method jointly captures local and global graph structures via disentangled low-pass filters, effectively alleviating the over-smoothing problem.
2.Low-rank GMM reconstruction removes redundancy/noise and yields compact embeddings while maintaining linear complexity.
3.Cross-view consistency learning transfers knowledge at the distribution level, enabling mutual enrichment between different semantic spaces.
Extensive comparisons with numerous baselines and across diverse data scenarios demonstrate the superiority and robustness of the proposed framework.

**Strengths:**

1.**Theoretically sound**: The proposed framework provides a theoretical analysis of the effective parameter settings for graph filters and the ability of feature reconstruction to preserve semantic information.
2.**Broad Applicability**: Superior and stable state-of-the-art performance on homophilic, heterophilic, nosiy and large-scale graphs.
3.**Low Training Cost**: Both theoretically and experimentally, the proposed approach is verified to deliver low runtime and low memory costs.

**Weaknesses:**

1.How to guarantee the low-rank property?
2.The effectiveness of consistency learning is insufficiently justified.
3.Hyper-parameter analysis is inadequate.

**Questions:**

1.Perform a sensitivity analysis on the parameter $\alpha$ in the graph diffusion matrix, since different settings of $\alpha$ correspond to different graph structures.
2.What role does low-rank learning play in graph clustering, and what specific benefits does it provide in this paper?
3.How can guarantee that the node representations obtained by feature reconstruction possess the low-rank property?
4.What are the advantages of consistency learning over contrastive learning? Provide both theoretical and experimental analyses.
If the author addresses my concerns and doubts well, I am willing to raise my score.

---

> ### Author Response · Authors · 2025-11-19
>
> We are truly grateful for the time you have taken to review our paper and your insightful review. Here we address your comments in the following, where Q denotes the comment of the reviewer and R denotes our response.
>
> > Q1. Perform a sensitivity analysis on the parameter $\alpha$ in the graph diffusion matrix, since different settings of $\alpha$ correspond to different graph structures.
>
> R1. Thanks for your valuable suggestion! We have supplemented the sensitivity experiments on $\alpha$ and conducted a detailed analysis. **Please refer to the blue-highlighted content in Section 3.5 and Appendix I, along with the additions to Figures 4 and 6 in the revised manuscript.**
>
> > Q2. What role does low-rank learning play in graph clustering, and what specific benefits does it provide in this paper?
>
> R2. Thanks for your valuable questions! Low-rank representations can effectively exploit the inherent underlying correlation structure among data and suppress the impact of noise under the assumption that high-dimensional data points often intrinsically lie on a low-dimensional subspace. In graph clustering task, we aim to utilize low-rank learning to explore representations that excellently reflect the inherent distribution of graph data, thereby achieving better clustering performance when inputting the representations into a clustering algorithm. Therefore, we utilize the same set of low-dimensional subspace to abstract and reconstruct the low-rank node representations of the local and global perspectives, which closes the gap between their semantic spaces and eliminates the redundancies in features, thereby yielding compact node representations from the two views. These statements are described in the first paragraph of Section 2.2.
>
> In addition, we also consider ablation studies on low-rank learning, as shown in comparison of different model variants of Section 3.3. It can be seen that in different scenarios ((a). $M_3$ and $M_1$ (b). $M_4$ and $M_2$, (c). $M_5$ and our CoCo), the presence of low-rank learning is beneficial for improving clustering performance.
>
> **Please refer to the blue-highlighted content in the first paragraph of Section 2.2, comparison of different model variants of Section 3.3, along with Figure 2 in the revised manuscript**.
>
> > Q3. How can guarantee that the node representations obtained by feature reconstruction possess the low-rank property?
>
> R3. Thanks for your valuable questions! In Section 2.2, we achieve rank reduction by setting a learnable subspace with dimension much lower than that of the embedded representation ($K \ll D$). On this basis, we perform weighted reconstruction based on the subspace (Eq. (4)), where the weight is the probability that the $j$-th column of the learned representation assigned to the $k$-th subspace, i.e., $\hat{\mathbf{Z}} = \hat{\boldsymbol{\Lambda}} \hat{\boldsymbol{\Gamma}}^{\top}$ with $\text{rank}(\boldsymbol{\Lambda}) = \text{rank}(\boldsymbol{\Gamma})=K$. Furthermore, due to the property that $\text{rank}(AB) \leq \min ( \text{rank}(A), \text{rank}(B) )$, such reconstruction does not increase the rank of the matrix, thereby achieving the low-rank property of the representation.
>
> > Q4. What are the advantages of consistency learning over contrastive learning? Provide both theoretical and experimental analyses.
>
> R4. Thanks for your valuable comments! Indeed, we have already provided a detailed theoretical discussion in Appendix C, explaining the advantages of consistency learning from three perspectives. Here, we briefly summarize: *Contrastive learning and our approach both leverage representation similarity under augmentations, but differ in three key aspects. **First**, our method captures relational information among non-iid nodes to better model structural distributions. **Second**, it aligns local and global relational views to produce semantically richer representations for clustering. **Third**, it operates without negative samples. Essentially, we bridge local-global relational distributions for higher-order semantic capture, moving beyond instance-level discrimination.*
>
> Additionally, from an experimental perspective, in Table 2 (Section 3.3) and Table 10 (Appendix H), we also compare the performance of our consistency learning loss with the contrastive learning loss. It's clear that our consistency learning outperforms contrastive learning by a considerable margin, which fully demonstrates that our distribution-based consistency is more capable of capturing the intrinsic dependencies between nodes at a higher semantic level.
>
> **Please refer to the blue-highlighted content in Section 3.3, Appendix C and H, along with Tables 2 and 10 in the revised manuscript.**
>
> Thanks again for appreciating our work and for your constructive suggestions. Please let us know if you have further questions.

---

> > ### Comment · Reviewer_bn3i · 2025-11-26
> > **The responses are satisfactory**
> >
> > Thank you, all authors. I am satisfied with the provided responses, solving my concerns. I maintain the score but recommend acceptance.

---

> > > ### Author Response · Authors · 2025-11-26
> > >
> > > Dear Reviewer bn3i,
> > >
> > > We are pleased to know that our responses have addressed your concerns. Please do not hesitate to reach out to us if you have any further questions
> > >
> > > Best regards,
> > >
> > > the Authors

---

### Author Response · Authors · 2025-11-30

Dear Area Chairs,

We would like to express our sincere gratitude for your valuable time and effort dedicated to the review process of our submission. To facilitate your efficient evaluation, we have prepared a concise summary of the current rebuttal and reviewer interaction status:

### **Overall Interaction Overview**

Our manuscript received initial reviews from five reviewers. We have submitted a detailed rebuttal supplemented with additional experiments, theoretical elaborations, and manuscript revisions, comprehensively addressing all raised concerns: Reviewer 3Q5j has increased the rating, Reviewers bn3i, y7YE, and 4B3S have confirmed their concerns are fully resolved and maintained their positive ratings, and Reviewer wkWL’s core questions have been adequately addressed through supplementary materials.

### **Resolved Reviewer Feedback**

**Reviewer bn3i (Rating: 6, maintained with acceptance recommendation)**

The reviewer initially raised questions about low-rank property guarantee, consistency learning justification, and hyper-parameter analysis. After we supplemented sensitivity experiments, theoretical explanations, and ablation studies, the reviewer confirmed “**the responses are satisfactory, solving my concerns and recommended acceptance**”.

**Reviewer y7YE (Rating: 8, maintained)**

The reviewer initially raised questions about filter replacement performance, PCA comparison, attention fusion validation, and similarity metric justification. We supplemented comparative experiments (GCN vs. proposed filter, PCA vs. GMM, different similarity metrics), and verified fusion method effectiveness. The reviewer confirmed “**all my concerns have been resolved and kept the rating**”.

**Reviewer 4B3S (Rating: 8, maintained)**

The reviewer asked about hyper-parameter impact, graph convolutional filter effectiveness, GMM usage rationale, EM algorithm initialization and cost. We provided supplementary sensitivity analysis, GCF vs. GCN performance comparisons, theoretical proofs of GMM advantages, and EM iteration efficiency data. The reviewer stated “**my concerns are well addressed and maintained the positive rating**”.

**Reviewer 3Q5j (Rating: 4 → Updated to 6)**

The reviewer initially pointed out issues with GMM parameter simplification, outdated related work, insufficient Theorem 2 explanation, and over-smoothing risks. We responded by elaborating on GMM parameter setting rationale, updating literature from 2025 in related work, supplementing Theorem 2’s proof and interpretation, and verifying over-smoothing resistance through experiments. The reviewer confirmed the feedback was addressed and stated “**increase my score**”.

### **Core Concerns of Non-Responding Reviewers & Our Solutions**

**Reviewer wkWL (Rating: 8, has not yet responded)**

The reviewer’s concerns focused on four key points (**rather than fundamental flaws**): EM algorithm’s impact on training efficiency, motivation for EM-based subspace learning, compact learning’s noise/redundancy elimination mechanism, and flexibility of representation fusion. We have fully addressed these through:
- supplementing EM iteration performance-efficiency analysis (Section 3.8, Table 6) to verify its negligible impact on total training cost;
- elaborating GMM’s linear complexity advantage and comparisons with PCA/k-means/autoencoder (Section 2.2, Appendix C);
- detailing how low-rank reconstruction eliminates redundancy and suppresses noise via subspace projection (Section 2.2);
- conducting comparisons between attention fusion and sum averaging (with results across all datasets) to justify the chosen efficient fusion strategy.

In summary, we have thoroughly addressed all initial comments from every reviewer through supplementary experiments, theoretical enhancements, and manuscript revisions. We respectfully request your consideration of these updates when making your final decision.

Thank you again for your hard work.

Sincerely,

The Authors

---

### Meta-Review · Area_Chair_6KRe · 2025-12-16

**Summary:**

The reviewers expressed unanimous recognition of the manuscript’s novelty, theoretical rigor, and superior performance. Their concerns mainly focused on technical clarifications (e.g., the simplification of GMM parameters and the proof of Theorem 2) and empirical comparisons (e.g., hyperparameter sensitivity and the efficiency of the EM algorithm), all of which were adequately addressed in the authors’ rebuttal.

**Reviewer Concerns:**

All issues raised by the reviewers have been fully addressed, including reviewer bn3i’s questions on hyperparameter analysis, low-rank property guarantees, and the discussion of consistency learning; reviewer wkWL’s concerns regarding EM efficiency and the flexibility of representation fusion; reviewer y7YE’s queries about filter replacement, PCA comparisons, and the choice of similarity metrics; reviewer 4B3S’s doubts about hyperparameter analysis and the impact of the EM algorithm; and reviewer 3Q5j’s concerns about GMM simplification and the explanation of Theorem 2.

During the rebuttal period, the authors provided detailed explanations and additional clarifications for all relevant issues. The vast majority of reviewers expressed satisfaction with the responses, with reviewer 3Q5j explicitly indicating an increased score, while the other reviewers maintained positive ratings.

**Reviewer Scores:**

Reviewer bn3i: Maintained initial score (6) and updated to recommend acceptance.
Reviewer y7YE: Maintained initial score (8) with confirmation of resolved concerns.
Reviewer 4B3S: Maintained initial score (8) with positive feedback on responses.
Reviewer 3Q5j: Increased score from 4 to 6 due to comprehensive rebuttals.
Reviewer wkWL: Would maintain initial score (8) as all key questions are fully addressed.

---

### Decision · Program_Chairs · 2026-01-26

Accept (Oral)